# TRPM7 and magnesium orchestrate human CD4 T-cell activation and differentiation

Anna Madlmayr[1,*] , Kilian Hoelting[2,*], Birgit Karner-Hoeger[1,3] , Dorothea Lewitz[2], Marius Weng[1], Severin Hacker[1], Julia Eder[1], Katharina Horner[1], Christine Schedlberger[1], Tanja Haider[1], Max Lechner[2], Michelle Duggan[1], Rylee Ross[4], F David Horgen[4], Markus Sperandio[5] , Alexander Dietrich[2] , Thomas Gudermann[2] , Susanna Zierler[1,2,3]

**T-lymphocyte activation is a crucial process in the regulation of innate and adaptive immune responses. The ion channel-kinase TRPM7, transient receptor potential cation channel subfamily M, member 7, has previously been implicated in cellular $Mg^{2+}$ homeostasis, proliferation, and immune cell modulation. Here, we show that pharmacological and genetic silencing of TRPM7 leads to diminished activation and influences signaling pathways that guide human $T_H17$ or $T_{reg}$ cell differentiation, following TCR-mediated stimulation. In primary human CD4 T cells and CRISPR-Cas9-engineered Jurkat T cells, inactivation or loss of TRPM7 led to distorted $Mg^{2+}$ homeostasis and $Ca^{2+}$ signaling, reduced NFAT translocation, decreased IL-2 secretion and altered $T_H$ cell differentiation. While the activation of primary human CD4 T cells, as well as in vitro polarization into pro-inflammatory $T_H17$ cells was critically dependent on TRPM7, the polarization of naïve CD4 T cells into FOXP3+ regulatory T cells was not. Taken together, these results highlight TRPM7 as molecular switch in lymphocyte activation and polarization. Thus, suggesting a therapeutic potential for TRPM7 in numerous T-cell mediated diseases.**

## Introduction

Immune cell function is essential for health and disease. Both innate and adaptive immune responses involve various cell types and are precisely regulated (Parenti et al, 2016; Walker, 2022). CD4 T lymphocytes are critically involved in both innate and adaptive immune responses (Parenti et al, 2016; Dong, 2021). Through different cellular subsets, CD4 T cells initiate adaptive immune responses against various kinds of pathogens. They have a crucial function in anti-cancer immunity, but also play a key role in the development of autoimmune diseases (Bonilla & Oettgen, 2010; Yatim & Lakkis, 2015; Abbas, 2019; Walker, 2022). Robust receptor-mediated cell activation, including various costimulatory signals, is crucial for lymphocyte function and ultimately leads to cell proliferation and differentiation into specific effector cell types (Bonilla & Oettgen, 2010; Heinzel et al, 2018; Martínez-Méndez et al, 2021). Accordingly, T-cell activation is the target of several established and emergent pharmacological strategies for immune modulation. Thus, gaining further insights into T-cell activation and the involvement of interaction partners is necessary to gain a better understanding of potential therapeutic targets.

Transient receptor potential cation channel subfamily M, member 7 (TRPM7), is a protein ubiquitously expressed in mammalian cells, showing high expression in lymphocytes (Beesetty et al, 2018; Krishnamoorthy et al, 2018). Embryonic development, thymopoiesis, and cellular proliferation critically rely on TRPM7 activity (Nadler et al, 2001; Jin et al, 2008; Beesetty et al, 2018; Nadolni et al, 2020). Expressing an ion channel in the plasma membrane, TRPM7 conducts divalent cations such as $Mg^{2+}$, $Ca^{2+}$, and $Zn^{2+}$, of which $Mg^{2+}$ ions are most relevant under physiological conditions (Nadler et al, 2001; Schmitz et al, 2003; Liang et al, 2022). Mutations in the *TRPM7* gene are associated with several clinical phenotypes in humans and mice. Most of the symptoms induced by TRPM7-mediated pathologies include macrothrombocytopenia and developmental delay (Stritt et al, 2016; Bosman et al, 2024), reduced $Mg^{2+}$ serum levels and signs of systemic inflammation, and can be ameliorated by $Mg^{2+}$ supplementation (Krishnamoorthy et al, 2018; Chubanov et al, 2024). Different studies have characterized TRPM7 as a key player of cellular $Mg^{2+}$ uptake (Cherepanova et al, 2016; Stritt et al, 2016; Hoeger et al, 2023), whereas other proteins proposed for this role, such as MAGT1 transporter, have lost scientific support (Li et al, 2011; Cherepanova et al, 2016; Ravell et al, 2020). Moreover, the TRPM7 ion channel domain is covalently linked to a cytosolic serine/threonine kinase domain (Nadler et al,

[1]Institute of Pharmacology, Faculty of Medicine, Johannes Kepler University Linz, Linz, Austria   [2]Walther Straub Institute of Pharmacology and Toxicology, Ludwig-Maximilians-Universität München, Munich, Germany   [3]Clinical Research Institute for Inflammation Medicine, Johannes Kepler University Linz, Linz, Austria   [4]Laboratory of Marine Biological Chemistry, Hawaiʻi Pacific University, Honolulu, HI, USA   [5]Institute of Cardiovascular Physiology and Pathophysiology, Biomedical Center, Ludwig-Maximilians-Universität München, München, Germany

Correspondence: susanna.zierler@jku.at
*Anna Madlmayr and Kilian Hoelting contributed equally to this work

2001; Schmitz et al, 2003; Liang et al, 2022). Different in vitro and native TRPM7 kinase substrates have been found, including myosin II, Annexin A1, phospholipase C gamma 2, SMAD2 and AKT (Dorovkov and Ryazanov, 2004; Clark et al, 2008; Romagnani et al, 2017; Hoeger et al, 2023). In recent years, important insights have been gained regarding the role of TRPM7 in mammalian immune cells. Absence of TRPM7 channel function has been linked to reduced store-operated $Ca^{2+}$ entry (SOCE) and proliferation arrest in DT40 chicken B cells and in a kinase-deficient mouse model (Sahni & Scharenberg, 2008; Faouzi et al, 2017; Beesetty et al, 2018; Krishnamoorthy et al, 2018). These mice showed reduced numbers of $T_H17$ cells and a concomitant protection from acute graft-versus-host disease (Romagnani et al, 2017). Here, we shed light on the role of TRPM7 in human T-lymphocyte activation and homeostasis. We demonstrate TRPM7 to be crucial for $Mg^{2+}$ allocation and $Ca^{2+}$ signaling, activation and proliferation of Jurkat T cells and primary CD4 T cells, as well as for subsequent effector functions including cytokine release and in vitro CD4 T-cell polarization.

# Results

## Maintenance of $Mg^{2+}$ homeostasis via TRPM7 is essential for Jurkat T-cell proliferation

Jurkat T cells are a well-characterized and a commonly used cell line to study T-lymphocyte function and signaling. We used this model to gain insight into the role of TRPM7 in T-cell signaling and activation. Applying CRISPR-Cas9 genome editing, we generated two clones of a novel TRPM7 KO Jurkat T-cell line harboring a genomic base pair insertion, which results in a frameshift in exon 4. The successful base pair insertion was confirmed through Sanger sequencing (Fig S1A). RT-qPCR and Western blot experiments showed *TRPM7* mRNA levels as well as TRPM7 protein expression to be reduced in both KO clones (Fig S1B and C). We were able to confirm the expected abolition of TRPM7 currents in these cells via whole-cell patch-clamp experiments, thereby functionally verifying the knockout (Figs 1A and B and S2A and B). TRPM7 KO clones showed a clear reduction of proliferation rates in standard Jurkat T-cell media and died within 5 d, whereas supplementation with 6 mM $MgCl_2$ restored normal proliferation and prevented cell death (Figs 1C and D and S2C and D). To further examine the nature of the TRPM7 KO T cells' need for $MgCl_2$ supplementation, we performed inductively coupled plasma mass spectrometry (ICP-MS), which revealed a reduction of cellular magnesium, $Mg^{2+}$, content in TRPM7 KO cells (Figs 1E and S2E). Culturing them in medium supplemented with 6 mM $MgCl_2$ restored intracellular $Mg^{2+}$ levels (Figs 1E and S2E). In parallel, we employed a pharmacological inhibitor of the TRPM7 channel, NS8593 (Chubanov et al, 2012), which similarly abolished TRPM7 currents in WT Jurkat T cells (Fig 1F and G). Culturing WT Jurkat T cells in the presence of NS8593 produced a similar effect as the TRPM7 KO. Treatment markedly reduced cell proliferation and viability within 5 d, with survival and proliferation being partially restored by supplementing extracellular $MgCl_2$ (Fig 1H

and I). Since NS8593 has been known to also inhibit SK2-channels in other cell types, we controlled for a potential SK2-dependent effect by employing the SK2-inhibitor Apamin, which did not influence TRPM7 currents in respective patch-clamp experiments (Fig S3A and B). Apamin likewise did not affect lymphocyte viability and growth (Fig S3C and D). To further rule out potential off-target effects of NS8593, we applied the TRPM7 inhibitor to Jurkat TRPM7 KO cells. Treatment of TRPM7 KO cells with NS8593 did not further affect viability and cell growth, suggesting no additional off-target effect of the inhibitor (Fig S3E and F). Similar to Jurkat TRPM7 KO clones, treatment with NS8593 also resulted in reduced cellular $Mg^{2+}$ levels, as analyzed by ICP-MS (Fig 1J), whereas $Mg^{2+}$ supplementation of the medium restored intracellular $Mg^{2+}$ levels (Fig 1J). In line with previous studies on TRPM7 (Zierler et al, 2011), these findings emphasize the importance of the ion channel for cell proliferation and $Mg^{2+}$ homeostasis in Jurkat T cells.

## TRPM7 channel activity is critical for Jurkat T-cell activation

Having tested the general functionality of our genetic and pharmacological models in Jurkat T cells, we aimed to decipher the role of TRPM7 in the activation process of human lymphocytes. Previously, TRPM7 was linked to altered store-operated $Ca^{2+}$ entry (SOCE) in DT40 chicken B lymphocytes (Faouzi et al, 2017). As an important early step in lymphocyte activation, we designed our experiments to first characterize the effects of TRPM7 in $Ca^{2+}$ signaling. Using Fura-2 as a ratiometric $Ca^{2+}$ indicator, we performed $Ca^{2+}$ imaging experiments comparing Jurkat TRPM7 WT and KO cells. Following depletion of the intracellular $Ca^{2+}$ stores using thapsigargin, TRPM7 KO cells exhibited a strongly reduced rise in cytosolic $Ca^{2+}$ concentration ($[Ca^{2+}]_i$) (Figs 2A and S2F), suggesting SOCE to be defective in Jurkat T cells lacking TRPM7. To quantify the amount of $Ca^{2+}$ present in the cytosol during the measurement, we calculated the area under the curve of the $Ca^{2+}$ traces (Figs 2B and S2G), showing a marked reduction of $[Ca^{2+}]_i$ in both KO T cells, indicating an early activation defect. The observed $Ca^{2+}$ signaling defect would likely affect subsequent transcription factor recruitment. Given that an increase in $[Ca^{2+}]_i$ is directly responsible for calcineurin-mediated dephosphorylation and subsequent nuclear translocation of NFAT molecules (Maguire et al, 2013; Lin et al, 2019; Park et al, 2020), we next tested $Ca^{2+}$-induced NFATc1 translocation. Basal levels of nuclear NFATc1 were comparable in WT and KO cells (Figs 2C and D and S2H and I). Again, using thapsigargin as stimulant, we were able to induce the translocation of NFATc1 into the nucleus in WT control cells. Thapsigargin-induced translocation was diminished in TRPM7 KO cells (Figs 2C, E, and F and S2H, J, and K). Having observed altered transcription factor recruitment, we assessed mRNA expression levels of *IL-2*, a well-known NFAT target gene (Maguire et al, 2013; Sakellariou et al, 2024). TRPM7 KO cells showed a remarkable reduction of *IL-2* mRNA (Fig 2G). One important feature of T-cell activation is the expression of activation markers on the cell surface, of which CD69 is robustly up-regulated in stimulated Jurkat T cells. 24 h after TCR activation, viable TRPM7 KO cells up-regulated CD69, however, to a significantly lower extent than TRPM7 WT cells

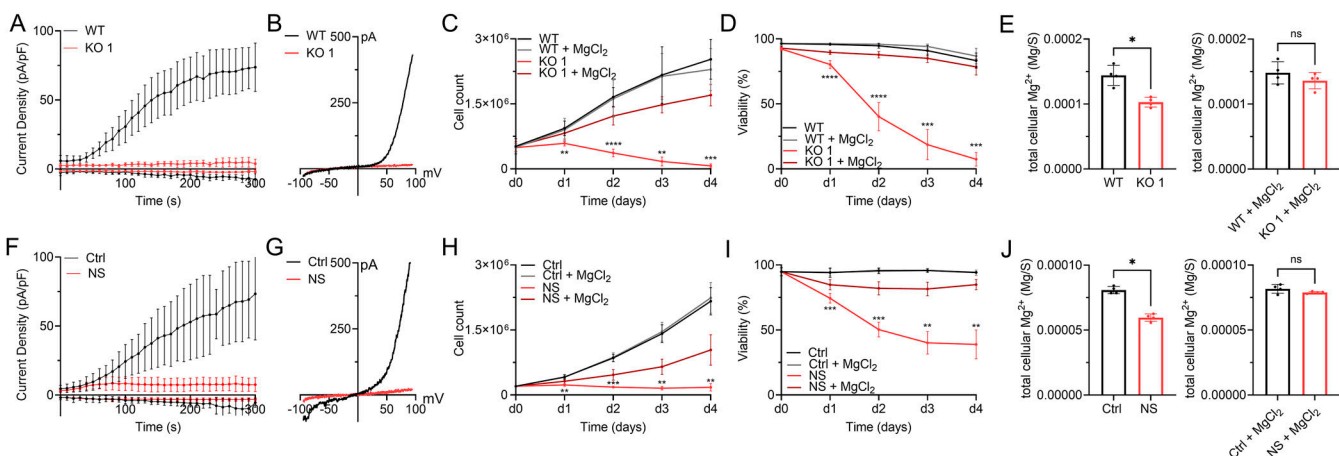

**Figure 1. Mg$^{2+}$ homeostasis via TRPM7 is essential for Jurkat T-cell proliferation.**
**(A, B)** TRPM7 current densities extracted at −80 and +80 mV, and (B) representative I/V relationships characteristic for TRPM7 channels in TRPM7 WT (black) and TRPM7 KO (red) Jurkat T-cell clones obtained via whole-cell patch clamp recordings in the absence of intracellular Mg$^{2+}$, n (WT) = 9; n (KO) = 10. **(C)** Cell counts and (D) viability of proliferating TRPM7 WT (black) and KO (red) Jurkat clones in medium, with or without supplementation of 6 mM MgCl$_2$, n = 3, measured in duplicates. **(E)** Cellular Mg$^{2+}$ content quantified by ICP-MS. TRPM7 WT (black) and KO (red) Jurkat T-cell clones cultured in regular media, with or without MgCl$_2$ supplementation for 18 h ahead of sampling, n = 4. **(F, G)** TRPM7 current densities extracted at −80 and +80 mV, and (G) representative TRPM7 I/V relationships of TRPM7 WT Jurkat T cells, treated with 30 $\mu$M NS8593 (NS, red) or DMSO control (Ctrl, black), obtained via whole-cell patch clamp recordings as in (A, B), n (Ctrl) = 6; n (NS) = 10. **(H)** Cell counts and (I) viability of proliferating Jurkat T cells in medium, with or without supplementation of 6 mM MgCl$_2$, treated with 30 $\mu$M NS8593 (NS, red) or DMSO control (Ctrl, black), n = 4. **(J)** Cellular Mg$^{2+}$ content quantified by ICP-MS. Jurkat WT cells, treated with 30 $\mu$M NS8593 (NS, red) or DMSO control (Ctrl, black) cultured in regular media with or without 6 mM MgCl$_2$ supplementation for 18 h ahead of sampling, n = 4. **(C, D, E, H, I, J)** Statistics: two-way ANOVA (C, D, H, I) or t test (E, J). *$P < 0.05$; **$P < 0.005$; ***$P < 0.0005$ and ****$P < 0.0001$. Data are mean ± SD.

(Figs 2H and I and S2L and M). Representative FACS plots for the gating strategy are shown in Fig S3I. We next performed the experiments with Jurkat T cells in the absence and presence of the TRPM7 channel inhibitor NS8593. Similar to the effect seen in the KO model, cells treated with the blocker exhibited a strong reduction of the [Ca$^{2+}$]$_i$ elevation (Fig 2J and K). We next tested Ca$^{2+}$-induced NFATc1 translocation. Basal levels of nuclear NFATc1 were comparable in untreated or TRPM7-inhibited cells (Fig 2L and M). Again, using thapsigargin as stimulant, we were able to induce the translocation of NFATc1 to the nucleus in WT control cells. Thapsigargin-induced translocation was diminished in cells treated with NS8593 (Fig 2L, N, and O). Similar to the TRPM7 KO model, cells treated with the TRPM7 inhibitor showed a remarkable reduction of *IL-2* mRNA (Fig 2P). Treatment of TRPM7 WT cells with NS8593 led to a significantly lower up-regulation of CD69 upon TCR activation, similar to what we observed in the TRMP7 KO model (Fig 2Q and R). In addition, we employed the TRPM7 inhibitor on Jurkat TRPM7 KO cells to apprehend potential off-target effects. NS8593 treatment of TRPM7 KO cells did not further affect Ca$^{2+}$ signaling (Fig S3G and H). We next checked up-regulation of CD69 in Jurkat cells upon CD3 stimulation. Notably, Apamin treatment did not influence CD69 expression (Fig S3J and K). TRPM7 KO cells up-regulated CD69 upon TCR stimulation, however, to a lesser extent than TRPM7 WT cells. This was not affected in TRPM7 KO cells after treatment with NS8593 (Fig S3I, L, and M). Overall, these data show a role of TRPM7 in modulating Ca$^{2+}$ signaling and downstream Ca$^{2+}$-dependent translocation of transcription factors and gene expression.

## TRPM7 inhibition alters Ca$^{2+}$ signaling and NFAT translocation in primary human CD4 T cells

Having validated NS8593 as an applicable pharmacological tool able to mimic the absence of TRPM7 protein in lymphocytes, we broadened the scope of the study to primary human CD4 T cells. Studying primary human lymphocytes instead of cell lines strongly increases the transferability of in vitro findings to immunological processes in human health and disease. CD4 T lymphocytes, isolated from healthy human PBMCs, were used to shed light on both naïve CD4 T cells (CD45RA$^+$) as well as total CD4 T cells (naïve and memory CD4 T cells). Isolated populations were validated by flow cytometry (Fig S4A and B). By whole-cell patch clamp, we were able to show functional channel expression of TRPM7 in naïve CD4 T cells. TRPM7 currents were absent after treatment with NS8593 (Fig 3A). Analogous to our Jurkat T-cell experiments, we characterized the Ca$^{2+}$-dependent activation cascade of primary CD4 T cells. We used antibodies against CD3 and CD28 to elicit TCR-dependent Ca$^{2+}$ signaling, which was analyzed by Fura-2-based Ca$^{2+}$ imaging. Applying stimulating antibodies to isolated naïve primary human CD4 T cells triggered a robust increase in [Ca$^{2+}$]$_i$ followed by oscillations of Ca$^{2+}$ concentration in a large subset of T cells (Fig 3B). Cells treated with the specific TRPM7 channel inhibitor NS8593 showed no reduction in basal Ca$^{2+}$ influx as well as in changes in intracellular Ca$^{2+}$ concentrations (Fig 3C–E) but had altered kinetics of [Ca$^{2+}$]$_i$ signals. Importantly, cytosolic Ca$^{2+}$ oscillations, which have been shown to be crucial for activation-induced gene expression (Dolmetsch et al, 1998), were absent

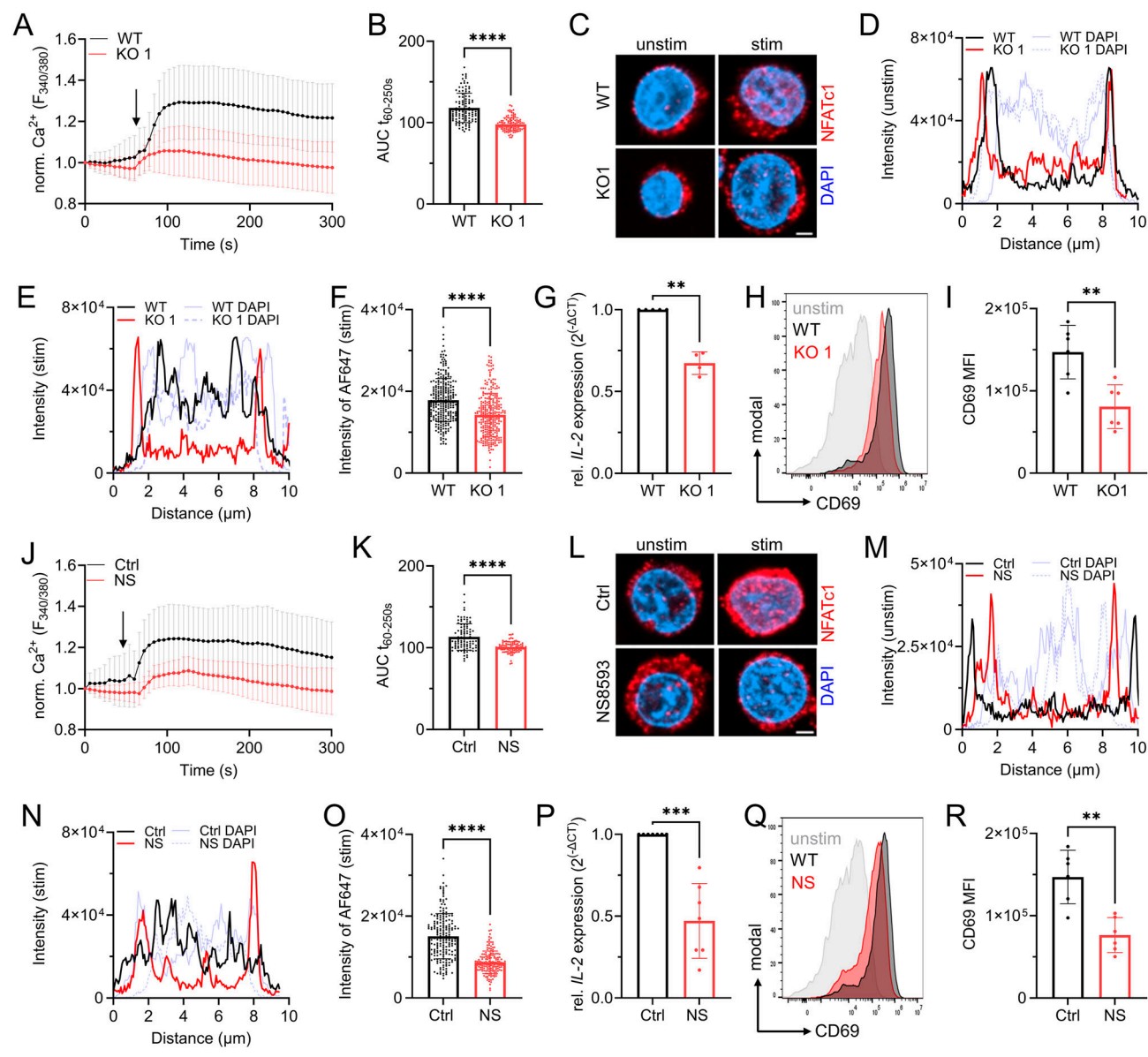

**Figure 2. TRPM7 is critical for Jurkat T-cell activation.**
**(A)** Fura-2 based imaging of cytosolic $Ca^{2+}$ concentration of Jurkat T cells. Passive store release was induced with 5 $\mu$M thapsigargin at the indicated time point (arrow) of WT (black) and TRPM7 KO (red) Jurkat T cells, n (WT) = 111; n (KO) = 113. **(B)** Quantification of the area under the curve (AUC) of respective curves shown in (A). **(C)** Representative immune-fluorescence images of the NFATc1 localization in TRPM7 WT and KO cells before (basal) and after 30 min stimulation (stim.) with 5 $\mu$M thapsigargin, scale bar = 2 $\mu$m. NFATc1 in red, DAPI in blue. **(D, E)** Representative intensity profiles of subcellular NFATc1 localization (NFATc1 in red, DAPI in blue) of Jurkat TRPM7 WT (black) and KO (red) cells in basal state (D) and upon 30 min passive store depletion induced with 5 $\mu$M thapsigargin (E). **(F)** Quantification of nuclear NFATc1 levels (corresponding to AF647 signal intensity) upon stimulation of TRPM7 WT (black) and KO (red) cells, n (WT) = 261; n (KO) = 279. **(G)** Relative *IL-2* mRNA expression levels of Jurkat TRPM7 WT (black) and KO (red) cells, n = 4. **(H)** Histograms and **(I)** quantification of up-regulated CD69 expression of Jurkat TRPM7 WT (black) and KO (red) cells after overnight stimulation with $\alpha$-CD3, n = 4–6. **(J)** Quantification of $Ca^{2+}$ signals of TRPM7 WT Jurkat T cells, treated with 30 $\mu$M NS8593 (NS, red) or DMSO control (Ctrl, black). Passive store release was induced with 5 $\mu$M thapsigargin at indicated time point (arrow), n (Ctrl) = 95; n (NS) = 94. **(K)** Quantification of the area under the curve (AUC) of respective $Ca^{2+}$ signals shown in (G). **(L)** Representative immune-fluorescence images of NFATc1 localization of cells treated with 30 $\mu$M NS8593 (NS, red) or DMSO control (Ctrl, black) before and after 30 min stimulation with 5 $\mu$M thapsigargin, scale bar = 2 $\mu$m. **(M, N)** Representative intensity profiles of subcellular NFATc1 localization (NFATc1 in red, DAPI in blue) of Jurkat TRPM7 WT (black) and KO (red) cells in basal state (M) and upon 30 min passive store depletion induced with 5 $\mu$M thapsigargin (N). **(O)** Quantification of nuclear NFATc1 levels upon stimulation of cells treated with 30 $\mu$M NS8593 (NS, red) or DMSO control (Ctrl, black), n (Ctrl) = 196; n (NS) = 195. **(P)** Relative *IL-2* mRNA expression levels of cells treated with 30 $\mu$M NS8593 (NS, red) or DMSO control (Ctrl, black), n = 7. **(Q)** Histograms and **(R)** quantification of up-regulated CD69 expression of cells treated with 30 $\mu$M NS8593 (NS, red) or DMSO control (Ctrl, black) after $\alpha$-CD3 stimulation, n = 6–7. **(B, D, E, F, H, J, K, M)** Statistics: *t* test (B, D, F, H, J, M) and Mann-Whitney *U* test (E, K). **$P < 0.005$; ***$P < 0.0005$; ****$P < 0.0001$ and n.s., not significant. Data are mean ± SD.

upon TRPM7 inhibition (Fig 3F). Studying the CD4$^+$CD25$^-$ effector T-cell population, also referred to as total CD4 T lymphocytes, displayed similar results. Using whole-cell patch clamp, we were able to show functional channel expression of TRPM7 in total CD4 T cells, which were absent upon treatment with NS8593 (Fig 3G). Applying the TRPM7 inhibitor NS8593, almost completely

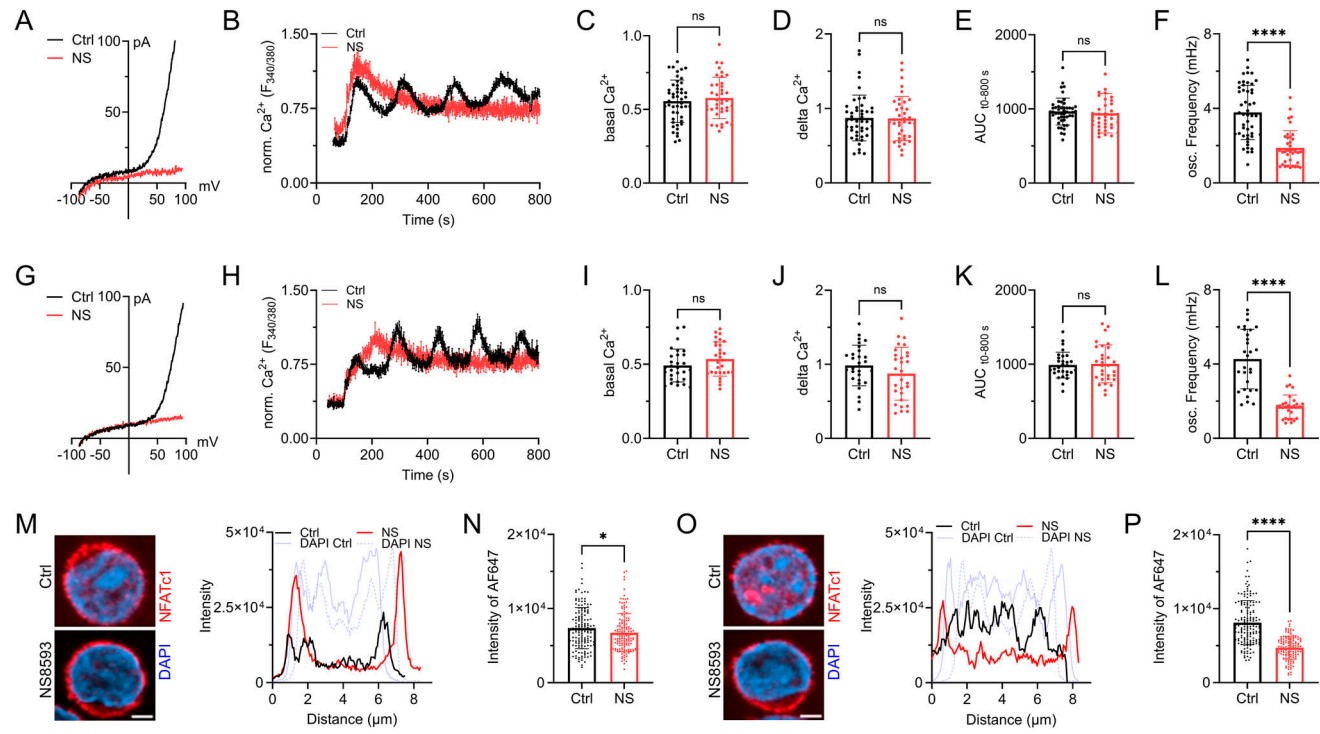

**Figure 3. TRPM7 inhibition alters Ca²⁺ signaling and NFAT translocation in primary human CD4 T cells.**
**(A)** Representative I/V relationships of TRPM7 channels in naïve CD4 T cells obtained via whole-cell patch clamp with Mg²⁺-free intracellular solution. Cells were treated with 30 μM NS8593 (NS, red) or DMSO control (Ctrl, black). **(B)** Representative single-cell traces of cytosolic Ca²⁺ concentrations of naïve CD4 T cells following α-CD3/α-CD28 stimulation. Cells were treated with 30 μM NS8593 (NS, red) or DMSO control (Ctrl, black) in saline (2 mM CaCl₂). **(C, D, E, F)** Quantification of Ca²⁺ signals of naïve CD4 T cells for (C) basal, (D) delta Ca²⁺, (E) AUC and (F) oscillation frequency, n = 39–48 cells. **(G)** Representative I/V relationships of TRPM7 channels in total CD4 T cells obtained via whole-cell patch clamp using Mg²⁺-free intracellular solution. Cells treated with 30 μM NS8593 (NS, red) or DMSO control (Ctrl, black). **(H)** Representative single-cell traces of cytosolic Ca²⁺ concentrations of total CD4 T cells following α-CD3/α-CD28 stimulation. Data obtained as in (B). Cells were treated with 30 μM NS8593 (NS, red) or DMSO control (Ctrl, black). **(I, J, K, L)** Quantification of Ca²⁺ signals of total CD4 T cells for (I) basal, (J) delta Ca²⁺, (K) AUC, and (L) oscillation frequency, n = 29–30 cells. **(M)** Representative immune-fluorescence images of NFATc1 localization (NFATc1 in red, DAPI in blue) and intensity profiles of subcellular NFATc1 distribution (Ctrl, black; NS, red; DAPI, light blue) of naïve CD4 T cells treated with 30 μM NS8593 (NS) or DMSO (Ctrl) upon 30 min stimulation with α-CD3/α-CD28, scale bar = 2 μm. **(N)** Quantification of nuclear NFATc1 levels upon 30 min α-CD3/α-CD28 stimulation of cells treated with 30 μM NS8593 (NS, red) or DMSO (Ctrl, black), n (Ctrl) = 149; n (NS) = 144. **(O)** Representative immune-fluorescence images of NFATc1 localization (NFATc1 in red, DAPI in blue) and intensity profiles of subcellular NFATc1 distribution (Ctrl, black; NS, red; DAPI, light blue) of total CD4 T cells treated with 30 μM NS8593 (NS) or DMSO (Ctrl) upon 30 min stimulation with α-CD3/α-CD28, scale bar = 2 μm. **(P)** Quantification of nuclear NFATc1 levels upon 30 min α-CD3/α-CD28 stimulation in presence of 30 μM NS8593 (NS, red) cells or DMSO control (Ctrl, black) or, n (Ctrl) = 155; n (NS) = 132. **(C, D, E, F, I, J, K, L, N, P)** Statistics: t test (C, D, E, F, I, J, K, L, N, P). *P < 0.05; ****P < 0.0001 and n.s., not significant. Data are mean ± SD.

eliminated Ca²⁺ oscillations in treated cells (Fig 3H–L). Application of the SK2 channel inhibitor Apamin as control for potential off-targets effects of NS8593, however, did not reduce Ca²⁺ oscillations (Fig S4C–E). With both the amount of Ca²⁺ as well as the characteristic Ca²⁺ oscillations known to be crucial for NFAT translocation (Maguire et al, 2013; Lin et al, 2019; Park et al, 2020), we proceeded by studying this process. We quantified NFATc1 residing in the nucleus after TCR-mediated stimulation in naïve and total CD4 T cells, as well as in cells treated with NS8593. Here, we saw in both cell subsets that TRPM7 inhibition resulted in reduced activation-dependent NFAT translocation (Fig 3M–P). This NS8593-induced defect in NFATc1-translocation highlights the importance of the Ca²⁺ oscillations, which were diminished in cells with TRPM7 blockade (Fig 3M–P). These results suggest an important role of TRPM7 in the early activation process of primary naïve and total CD4 T cells with large implications on activation-dependent gene expression.

**TRPM7 and Mg²⁺ control activation and proliferation of primary human CD4 T cells**

As transcription factor recruitment is crucial for IL-2 expression (Maguire et al, 2013; Sakellariou et al, 2024), we next investigated the stimulation-dependent release of this autocrine and paracrine cytokine of CD4 T cells. After 48 h of stimulation, control cells had secreted significantly more IL-2 into the supernatant than cells treated with NS8593. This effect could be partially rescued by MgCl₂ supplementation (Fig 4A). We next investigated activation-induced protein expression. Up-regulation of CD69 and CD25 are important hallmarks of T-cell activation, both being physiologically significant and well-studied (Nisnboym et al, 2023; Peng et al, 2023; Poloni et al, 2023). In response to α-CD3/α-CD28 stimulation, both activation markers were up-regulated in primary CD4 T cells, shown by representative FACS plots (Fig 4B–E) and corresponding gating strategy in Fig S4F. Treatment with NS8593 markedly reduced up-regulation of CD69 and CD25, an effect that could be reverted with

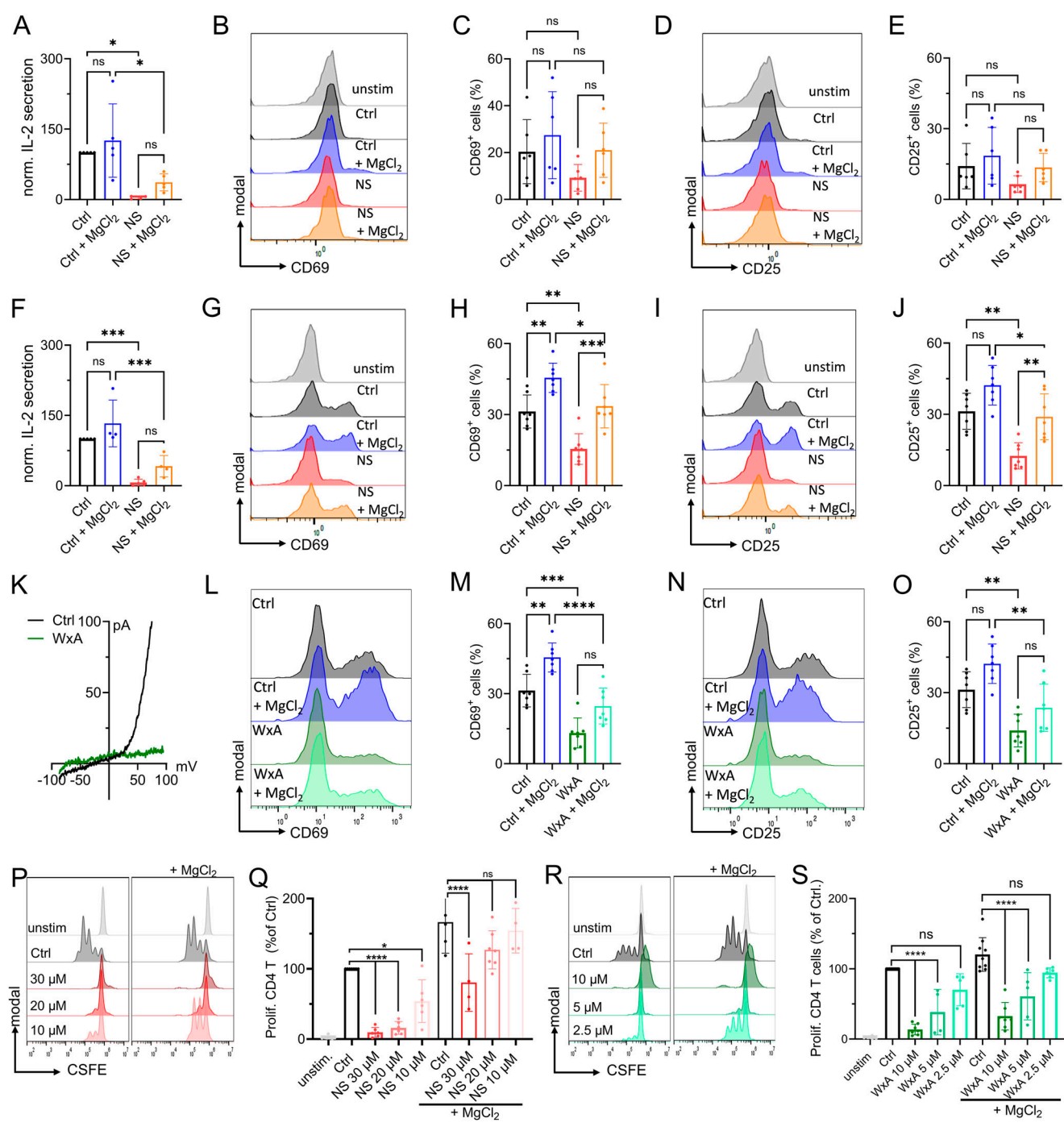

**Figure 4. TRPM7 and Mg²⁺ control activation and proliferation of primary human CD4 T cells.**
(A) IL-2 quantification in supernatant of naïve CD4 T cells 48 h after $\alpha$-CD3/$\alpha$-CD28 stimulation, n = 4–5. (B, C, D, E) Histograms and quantification of up-regulated activation markers CD69 (B, C) and CD25 (D, E) in naïve CD4 T lymphocytes 48 h after stimulation. Cells were treated with 30 $\mu$M NS8593 or DMSO control, both with (Ctrl, blue; NS, orange) and without (Ctrl, black; NS, red) supplementation of 6 mM MgCl$_2$. (F) IL-2 quantification in supernatant of total CD4 T cells 48 h after $\alpha$-CD3/$\alpha$-CD28 stimulation or cells treated with 30 $\mu$M NS8593 or DMSO control, both with (Ctrl, blue; NS, orange) and without (Ctrl, black; NS, red) supplementation of 6 mM MgCl$_2$, n = 4–5. (G, H, I, J) Histograms and quantification of up-regulated activation markers CD69 (G, H) and CD25 (I, J) in total CD4 T lymphocytes 48 h after stimulation. Cells treated with either 30 $\mu$M NS8593 or DMSO control, both with (Ctrl, blue; NS, orange) and without (Ctrl, black; NS, red) supplementation of 6 mM MgCl$_2$. (K) Representative TRPM7 I/V relationships of total CD4 T cells obtained via whole-cell patch clamp with Mg²⁺-free intracellular solution. Cells were treated with 10 $\mu$M Waixenicin A (WxA, green) or EtOH control (Ctrl, black). (L, M, N, O) Histograms and quantification of up-regulated activation markers CD69 (L, M) and CD25 (N, O) in total CD4 T lymphocytes 48 h after stimulation. Cells treated with 10 $\mu$M Waixenicin A or EtOH control, both with (Ctrl, blue; WxA, light green) and without (Ctrl, black; WxA, green) supplementation of 6 mM MgCl$_2$, n = 7. (P) Representative histograms of dose-dependent proliferation (CSFE dye dilution) of total CD4 T cells in presence of various NS8593 concentrations, with (right) and without (left) supplementation of 6 mM MgCl$_2$. Cells gated on T cell population, single cells and CD4⁺ T cells. Color code as in (Q). (Q) Respective quantification of NS8593 dose-dependent proliferation of total CD4 T cells, with and without supplementation of 6 mM MgCl$_2$, corresponding to (P), n =

MgCl$_2$ supplementation. MgCl$_2$ supplementation also increased the up-regulation of activation markers in control cells, underlining the importance of Mg$^{2+}$ in T-cell activation (Fig 4B–E). Similar as the naïve CD4 T cell subset, also the total CD4 population secreted large amounts of IL-2 upon TCR stimulation, which was significantly reduced upon TRPM7 inhibition. This effect could be partially rescued by MgCl$_2$ supplementation (Fig 4F). Treatment with NS8593 negatively affected up-regulation of both activation markers CD69 and CD25, which was partially rescuable with MgCl$_2$ supplementation (Fig 4G–J). Although TCR-mediated CD69 and CD25 up-regulation was, as expected, less pronounced in naïve T cells compared with the total CD4 T cells, inhibition of TRPM7 yielded similar effects in both cell populations (Fig 4B–E and G–J). Titration of inhibitor NS8593 showed a dose-dependent reduction of CD69 and CD25 up-regulation in total CD4 T cells (Fig S4G and H). Again, Apamin was employed to check for potential off-target effects of NS8593. Up-regulation of both CD69 and CD25 was not affected and comparable between cells treated with Apamin and vehicle controls (Fig S4I and J). To improve methodic robustness, we repeated our experiments with another known specific TRPM7 channel inhibitor, Waixenicin A (Zierler et al, 2011). By whole-cell patch clamp, we were able to confirm blockade of TRPM7 currents upon pharmacological treatment with Waixenicin A (Fig 4K). Both inhibitors yielded a similar up-regulation of CD69 and CD25 in these cells upon TCR-mediated stimulation (Fig 4L–O), which strongly supports a TRPM7-dependent effect. In proliferation experiments following α-CD3/α-CD28 stimulation, we observed robust proliferation of the activated CD4 T cells within 5 d. Treatment with NS8593 reduced cell proliferation (Fig 4P and Q). This effect was dose-dependent and could be partially reversed by supplementation of MgCl$_2$ (Figs 4P and Q and S4K and L), whereas Apamin treatment had no effect on CD4 T-cell proliferation (Fig S4M and N). Treatment of CD4 T cells with Waixenicin A had a similar effect on T-cell proliferation. Similar to NS8593, also Waixenicin A led to a dose-dependent inhibition of CD4 T-cell proliferation, which was not rescuable by MgCl$_2$ supplementation (Figs 4R and S and S4O). In summary, TRPM7 affects transcription factor recruitment, IL-2 secretion, and the up-regulation of activation-dependent surface markers in both, naïve and total CD4 T cells. This subsequently influences CD4 T-cell proliferation.

### TRPM7 interacts with and affects AKT1 and SMAD2 signaling in CD4 T cells

An important hallmark of adaptive immunity and a consequence of successful T-cell activation are increased proliferation, clonal expansion, and differentiation. Mendu et al., recently linked TRPM7 with enhanced development of regulatory T (T$_{reg}$) cells in a TRPM7 thymic-specific knockout mouse model (Mendu et al, 2020). In line, we have shown in Romagnani et al., that T$_H$17 development is negatively impacted by TRPM7 kinase in a genetic TRPM7 kinase-deficient mouse model (Romagnani et al, 2017). Thus, we investigated the role of TRPM7 in the in vitro differentiation of naïve human CD4 T cells into iT$_{reg}$ cells and counterbalancing iT$_H$17 cells. We first investigated related signaling mechanisms. The mTOR inhibitor rapamycin is a widely used promotor of iT$_{reg}$ cells and FOXP3 stability (Schmidt et al, 2016). We have recently shown that AKT is a direct target of TRPM7 kinase in CML cells (Hoeger et al, 2023) that could be similarly affected in T cells. We thus investigated a possible molecular involvement of AKT-mTOR inhibition upon TRPM7 inhibition in CD4 T cells. Indeed, we found pAKT signals in α-CD3/α-CD28-stimulated CD4 T cells to be reduced upon NS8593 treatment (Fig 5A and B). Importantly, total AKT1 levels were unaltered upon TRPM7 inhibition (Fig S5A and B). By proximity ligation, we were able to show a molecular interaction between AKT1 and TRPM7 upon TCR stimulation in CD4 T cells, which was almost completely abolished upon TRPM7 inhibition (Fig 5C and D), indicating a potential link to FOXP3-expressing T$_{reg}$ cells. Previously, in a TRPM7 kinase-deficient mouse model, we showed that TRPM7 engages with TGF-β-mediated induction of T$_H$17 cells, by phosphorylating SMAD2 (Romagnani et al, 2017). In accordance, we observed reduced phosphorylation of SMAD2 in human CD4 T cells treated with the TRPM7 inhibitor NS8593, compared with control (Fig 5E–H). SMAD2 phosphorylation in presence of 6 mM MgCl$_2$, led to a slightly, however, not significant increase of pSMAD2 signals (Fig 5I–L). Notably, total SMAD2 protein levels were unaltered in both treatment conditions (Fig S5C–F). Confirming what we have observed in the murine kinase-deficient model (Romagnani et al, 2017), by proximity ligation we identified a molecular interaction between TRPM7 and SMAD2 protein in the human CD4 T cells (Fig 5M). The interaction of TRPM7 and SMAD2 was completely abolished upon TRPM7 blockade (Fig 5M and N), suggesting a similar effect on human CD4 T-cell polarization.

Aside of TGF-β-SMAD2 signaling, the signal transducer and activator of transcription 3 (STAT3) is a key player in IL-6-mediated T$_H$17 differentiation by mediating pro-inflammatory cytokine production (Hirahara et al, 2010; Betts et al, 2014). STAT3 activity is tightly regulated by two critical phosphorylation events at Tyr705 via Janus tyrosine kinase members upon TCR stimulation, and the noncanonical Ser727 via the MAPK and JNK family (Zhu et al, 2021; Cheung et al, 2022; Qin et al, 2024). To investigate a possible interaction of the Ser/Thr kinase TRPM7 with STAT3 signaling in T$_H$17 polarization, we tested both STAT3 phosphorylation sites via IL-6 and TCR (α-CD3/α-CD28) stimulation in absence or presence of TRPM7 inhibitor. TCR stimulation was shown to positively induce phosphorylation of STAT3 Ser727 in T cells, which might play a role in promoting T$_H$17 (Ng & Cantrell, 1997; Cheung et al, 2022). We did not detect differences in pSTAT3 Tyr705 signaling upon TRPM7 inhibition in IL-6-stimulated cells (Fig S5G–J). These findings are in accordance with what we have seen in TRPM7 kinase-deficient murine T cells (Romagnani et al, 2017). As expected, STAT3 Tyr705 was not

4–7. **(R)** Representative histograms of dose-dependent proliferation (CSFE dye dilution) of total CD4 T cells in presence of various Waixenicin A concentrations, with (right) and without (left) supplementation of 6 mM MgCl$_2$. Cells gated on T cell population, single cells and CD4$^+$ T cells. Color code as in (S). **(S)** Respective quantification of Waixenicin A dose-dependent proliferation of total CD4 T cells, with and without supplementation of 6 mM MgCl$_2$, corresponding to (S), n = 4–8. **(A, C, E, F, H, J, M, O, Q, S)** Statistics: one-way ANOVA (A, C, E, F, H, J, M, O, Q, S). *$P < 0.05$; **$P < 0.005$; ***$P < 0.0005$; ****$P < 0.0001$ and n.s., not significant. Data are mean ± SD.

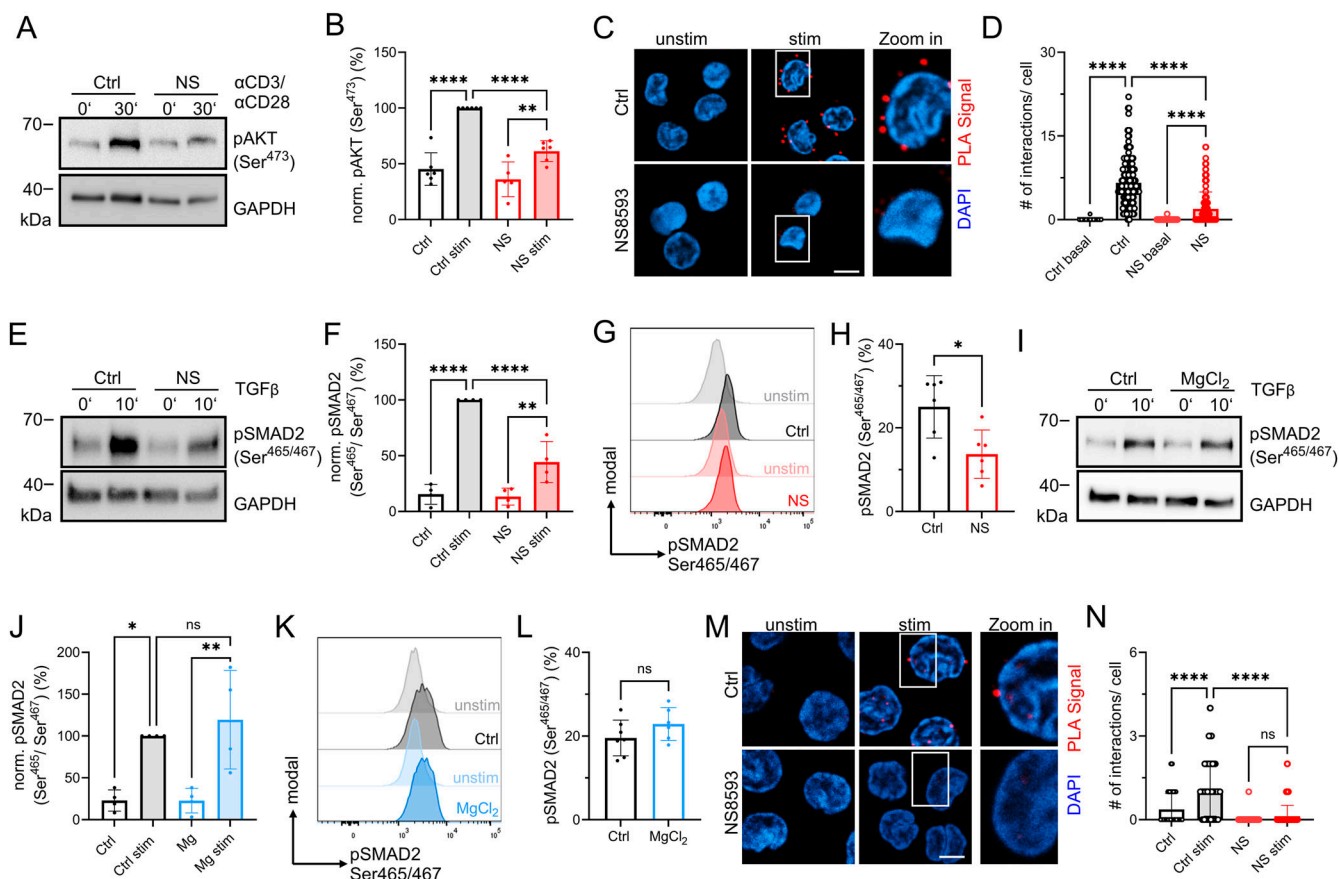

**Figure 5. TRPM7 interacts with and affects AKT1 and SMAD2 signaling in human CD4 T cells.**
**(A)** Representative Western blot of pAKT Ser473 signals in CD4 T cells in presence of 30 $\mu$M NS8593 (NS) or DMSO (Ctrl) in resting state and after 30 min $\alpha$-CD3/$\alpha$-CD28 stimulation. **(B)** Respective quantification of pAKT Ser473 signals from blots shown in (A), n = 6. **(C)** Representative immune-fluorescence images of AKT1-TRPM7 proximity-ligation interaction in CD4 T cells in resting state or after stimulation with $\alpha$-CD3/$\alpha$-CD28 for 30 min. AKT1-TRPM7 interactions are visible as red dots, DAPI in blue, scale bar = 5 $\mu$m. **(D)** Respective quantification of PLA signals per cell upon $\alpha$-CD3/$\alpha$-CD28 stimulation as shown in (C). Cells were treated with 30 $\mu$M NS8593 (NS, red) or DMSO control (Ctrl, black), n (Ctrl) = 173; n (NS) = 115. **(E)** Representative Western blot of pSMAD2 Ser465/Ser467 signals in CD4 T cells in presence of 30 $\mu$M NS8593 (NS) or DMSO control (Ctrl) in resting state and after 10 min TGF-$\beta$ stimulation. **(F)** Respective quantification of pSMAD2 Ser465/Ser467 signals from blots shown in E, n = 6. **(G)** Histogram and (H) quantification of pSMAD2 Ser465/Ser467 signals in CD4 T cells after 15 min TGF-$\beta$ stimulation. Cells treated with 30 $\mu$M NS8593 (NS) or DMSO control (Ctrl), unstimulated Ctrl shown in light gray, n = 6. **(I)** Representative Western blot of pSMAD2 Ser465/Ser467 signals in CD4 T cells in presence of 6 mM MgCl$_2$ (MgCl$_2$) or H$_2$O control (Ctrl) in resting state and after 10 min TGF-$\beta$ stimulation. **(J)** Respective quantification of pSMAD2 Ser465/Ser467 signals from blots shown in (I), n = 4. **(K)** Histogram and (L) quantification of pSMAD2 Ser465/Ser467 signals in CD4 T cells after 15 min TGF-$\beta$ stimulation. Cells treated with 6 mM MgCl$_2$ (MgCl$_2$, blue) or H$_2$O control (Ctrl, black), unstimulated Ctrl shown in light gray, n = 6–7. **(M)** Representative immune-fluorescence images of SMAD2-TRPM7 proximity-ligation interaction in CD4 T cells in resting state or after stimulation with TGF-$\beta$ for 10 min. SMAD2-TRPM7 interactions are visible as red dots, DAPI in blue, scale bar = 5 $\mu$m. **(N)** Respective quantification of PLA signals per cell upon TGF-$\beta$ stimulation as shown in (M). Cells were treated with 30 $\mu$M NS8593 (NS, red) or DMSO control (Ctrl, black), n (Ctrl) = 327; n (NS) = 284. **(B, D, F, H, J, L, N)** Statistics: one-way ANOVA (B, F, J) and t test (D, F, H, L, N). *$P$ < 0.05; **$P$ < 0.005; ****$P$ < 0.0001 and n.s., not significant. Data are mean ± SD.

detectable in TCR-stimulated cells (Fig S5K and L). Interestingly, when we tested phosphorylation of STAT3 Ser727, we similarly did not observe significant changes upon TRPM7 blockade upon IL-6 and TRC stimulation, respectively (Fig S5M–P). Of note, we only observed positive signals for TCR-induced STAT3 Ser727 phosphorylation by Western blot, not by flow cytometry, indicating an overall weak response via this stimulus (Fig S5Q–T). Hence, STAT3 only seems to play a minor role in TRPM7-mediated induction of T$_H$17 cells. STAT3 directly competes with STAT5 for DNA binding (Burchill et al, 2007), which induces FOXP3 downstream of IL-2 signaling, driving T$_{reg}$ cell development and blocking T$_H$17 differentiation (Burchill et al, 2007; Betts et al, 2017; Harrington et al, 2023). We therefore also tested STAT5 Tyr694

phosphorylation in absence or presence of TRPM7 inhibitor, comparing TCR and IL-2 stimulation. Similar to our data on STAT3, we did not observe differences in pSTAT5 Tyr694 signaling upon TRPM7 blockade (Fig S5U–W).

### TRPM7 inhibition during naïve human CD4 T-cell polarization preserves T$_{reg}$ and dampens T$_H$17 cell signatures

We next performed T-cell differentiation experiments toward the immunosuppressive FOXP3-expressing regulatory T cells as well as towards pro-inflammatory ROR$\gamma$t-expressing T$_H$17 cells upon TRPM7 inhibition (Fig 6A). Readout for successful iT$_{reg}$ cell differentiation was assessed by quantification of CD45RA$^+$ status, up-

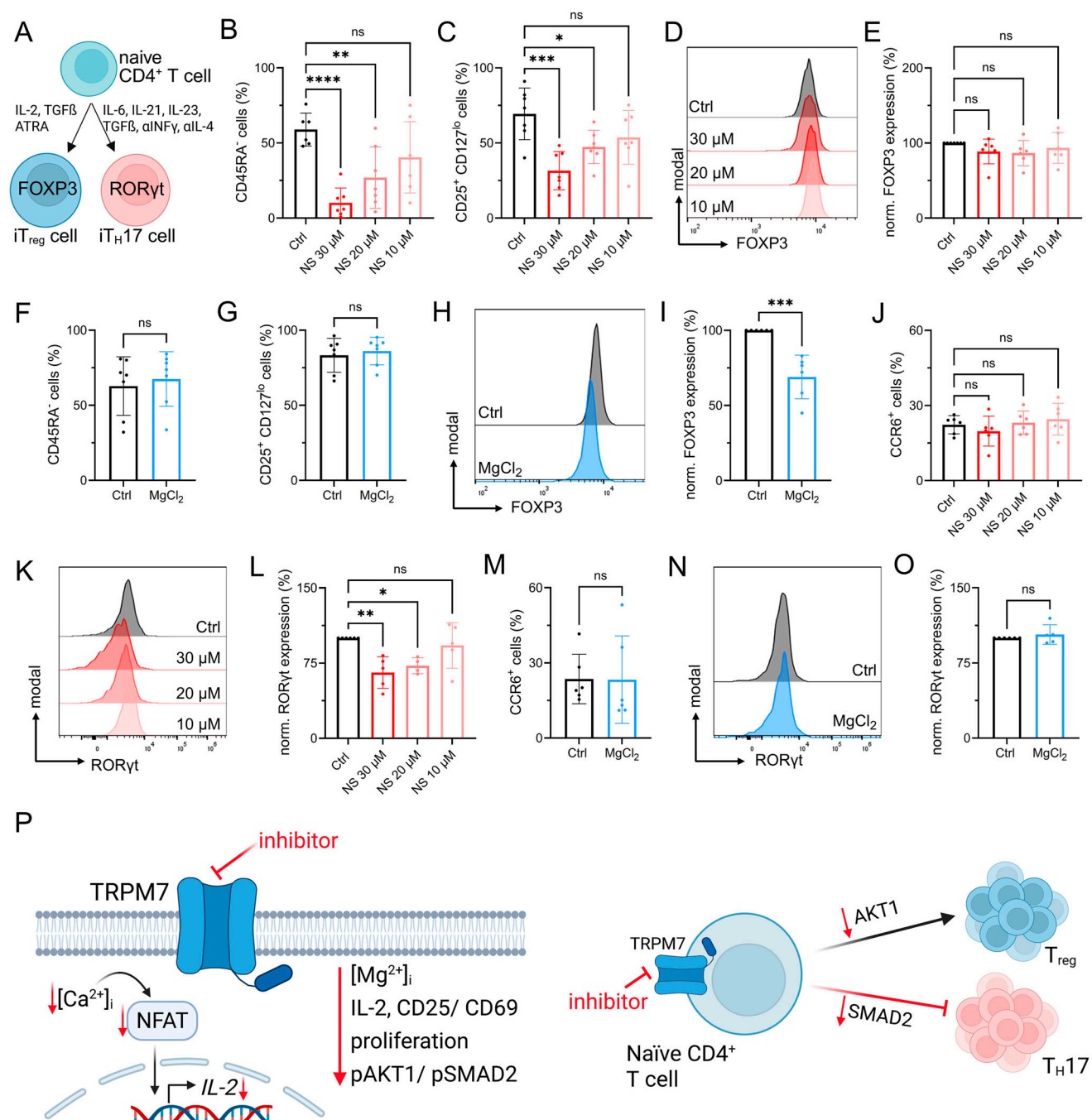

**Figure 6. TRPM7 inhibition influences naïve human CD4 T-cell differentiation in vitro by preserving T$_{reg}$ and dampening T$_H$17 signatures.**
**(A)** Schematic description of naïve CD4 T-cell differentiation towards FOXP3-expressing regulatory T cells and RORγt-expressing T$_H$17 cells, including respective cytokine polarization milieus. **(B, C)** Percentages of CD45RA$^-$ cells and (C) CD25$^+$CD127$^{lo}$ cells upon polarization of naïve CD4 T cells toward iT$_{reg}$ cells in various NS8593 concentrations (red) compared with DMSO control (Ctrl, black), n = 6–7. **(D, E)** Representative FACS histograms and (E) quantification of FOXP3 expression levels of CD25$^+$CD127$^{lo}$ iT$_{reg}$ cells upon 6 d polarization of naïve CD4 T cells in presence of various NS8593 concentrations (red) or DMSO control (Ctrl, black), n = 6–7. **(F, G)** Percentages of CD45RA$^-$ cells and (G) CD25$^+$CD127$^{lo}$ cells upon polarization of naïve CD4 T cells toward iT$_{reg}$ cells in presence of 6 mM MgCl$_2$ (MgCl$_2$, blue) compared with H$_2$O control (Ctrl, black), n = 7. **(H, I)** Representative FACS histograms and (I) quantification of FOXP3 expression levels of CD25$^+$CD127$^{lo}$ iT$_{reg}$ cells upon 6 d polarization of naïve CD4 T cells in presence of 6 mM MgCl$_2$ (MgCl$_2$, blue) compared with H$_2$O control (Ctrl, black), n = 6. **(J)** Percentages of CCR6$^+$ cells upon polarization of naïve CD4 T cells towards iT$_H$17 cells in presence of various NS8593 concentrations (red) compared with DMSO control (Ctrl, black), n = 6. **(K, L)** Representative FACS histograms and (L) quantification of RORγt expression levels of CCR6$^+$ iT$_H$17 cells upon 6 d polarization of naïve CD4 T cells in presence of various NS8593 concentrations (red) or DMSO control (Ctrl, black), n = 4–6. **(M)** Percentages of CCR6$^+$ cells upon polarization of naïve CD4 T cells towards iT$_H$17 cells in presence of 6 mM MgCl$_2$ (MgCl$_2$, blue) compared with H$_2$O control (Ctrl, black), n = 6. **(N, O)** Representative FACS histograms and (O) quantification of RORγt expression levels of CCR6$^+$ iT$_H$17 cells upon 6 d polarization of naïve CD4 T cells in presence of 6 mM MgCl$_2$ (MgCl$_2$, blue) compared with H$_2$O control (Ctrl, black), n = 5. **(P)** Graphical summary of TRPM7-(in)dependent T-cell activation and differentiation towards iT$_{reg}$ and iT$_H$17 cells. Pharmacological blockade of TRPM7 reduces intracellular Mg$^{2+}$ levels, leads to reduced Ca$^{2+}$ signaling and results in reduced IL-2 secretion, impaired up-regulation of T-cell activation markers CD69 and CD25, and diminished proliferation upon TCR stimulus (left).

regulation of CD25 in combination with reduction of CD127 expression (CD25$^+$CD127$^{lo}$) in CD45RA$^-$ cells, and subsequent FOXP3 expression levels. TRPM7 inhibition with NS8593 during polarization of naïve CD4 T cells toward T$_{reg}$ cells led to a dose-dependent reduction in T-cell proliferation and differentiation, as treated cells showed reduced percentages of CD45RA$^-$ status (Fig 6B). In accordance, we observed a dose-dependent reduction of CD25$^+$CD127$^{lo}$ cells (Fig 6C). Nevertheless, in line with our in vitro data on reduced AKT phosphorylation, in the presence of the TRPM7 inhibitor NS8593, we observed a preserved FOXP3 expression in the CD45RA$^-$CD25$^+$CD127$^{lo}$ iT$_{reg}$ cells (Figs 6D and E and S6A). Treatment with Apamin had little impact on FOXP3 expression (Fig S6B). Cells treated with the afore employed TRPM7 inhibitor Waixenicin A mainly did not activate and maintained their CD45RA$^+$ status, and as a consequence, CD25$^+$CD127$^{lo}$ cells were not detectable in the low CD45RA$^-$ populations (Fig S6C–F). These results reflect our proliferation experiments using Waixenicin A. Further, in contrast to data obtained with NS8593, our experiments revealed a negative effect of Mg$^{2+}$ on iT$_{reg}$ cell polarization. Whereas the CD45RA$^+$ status as well as percentages of CD25$^+$CD127$^{lo}$ cells were comparable between control cells and cells treated with 6 mM MgCl$_2$ (Fig 6F and G), FOXP3 expression was significantly reduced upon MgCl$_2$ addition (Fig 6H and I). Vice versa, we investigated the effect of TRPM7 inhibition on polarization of T$_H$17 cells (iT$_H$17) in vitro (Fig S7A). Treatment with TRPM7 inhibitor NS8593 had a dose-dependent effect on cell viability (Fig S7B) but had no effect on CCR6 expression (Fig 6J). Importantly, inhibition of TRPM7 with NS8593 led to a dose-dependent reduction in RORɣt expression in CCR6$^+$ iT$_H$17 cells (Fig 6K and L). These results are in line with what we have previously observed in a TRPM7 kinase-deficient mouse model (Romagnani et al, 2017). When we polarized naïve human CD4 T cells into iT$_H$17 cells in presence of 6 mM MgCl$_2$, we observed no effect on cellular viability (Fig S7C), or on CCR6 or RORɣt expression compared with control (Fig 6M–O). Again, the control compound Apamin had no impact on cellular viability, CCR6 or RORɣt expression (Fig S7D–F). Altogether, these results outline the essential function of TRPM7 in human T$_H$17 differentiation. In summary, our findings highlight TRPM7 as fundamental key player in human T-cell activation and signaling and point toward a modulatory role of TRPM7 in iT$_{reg}$ and iT$_H$17 cell differentiation in vitro, most likely by controlling Mg$^{2+}$ homeostasis and related kinase signaling events, as summarized in Fig 6P.

## Discussion

T-lymphocyte activation is a crucial process with implications for the whole immune system (Rock et al, 2011; Sakaguchi et al, 2020; Walker 2022). While pro-inflammatory T$_H$17 cells drive inflammation and autoimmune diseases, regulatory T cells (T$_{reg}$) are important to regulate and suppress immune responses and to maintain immunological self-tolerance (Paradowska-Gorycka et al, 2020; Wang et al, 2023). Both cell types play a role in maintaining immune homeostasis, in development of autoimmune diseases or graft-versus-host disease in patients with organ transplants (Haxhinasto et al, 2008; Sakaguchi et al, 2020; He et al, 2024). The ability to pharmacologically influence and reduce T-cell activation and differentiation is a primary therapeutic strategy for many autoimmune defects (Rock et al, 2011; Sakaguchi et al, 2020; Walker 2022). Gaining further insight into these complex activation processes is needed to unravel the pathogenesis and treatment options for a multitude of immunopathologies. We, here, conducted the first functional study on TRPM7 activity in primary human T lymphocytes. While TRPM7 had already been linked to numerous aspects of T-cell activation in different mouse models and cell lines (Romagnani et al, 2017; Beesetty et al, 2018; Mellott et al, 2020), we now characterize TRPM7 as an important and potentially druggable player of human lymphocyte activation, proliferation and most interestingly, polarization. We used pharmacological inhibitors to study the role of TRPM7 in primary human CD4 T cells. The risk of unspecific pharmacologic effects was mitigated by validating our approach by comparison with a genetic TRPM7 knockout model in Jurkat T cells, and by using two different, well-established TRPM7 inhibitors (NS8593, Waixenicin A) in key experiments. Rescue experiments by supplementation with MgCl$_2$ further underline the importance of TRPM7 activity for CD4 T-cell function. Mg$^{2+}$ serves as essential component of almost all cellular functions, providing structural integrity to membranes, ribonucleotides, serving as bioenergetic driver and as crucial component of kinase functions in the form of Mg:ATP, constituting a cofactor for numerous enzymes and thus playing a considerable role in almost all cellular signaling events (Wolf Federica & Achille, 2003; Romani, 2011; Baaij et al, 2015). Which proteins facilitate cellular Mg$^{2+}$ uptake in T cells, and whether TRPM7 is one of them, has been a contentious issue in the past (Li et al, 2011; Stangherlin and O'Neill, 2018; Castiglioni et al, 2023). MAGT1, long believed to be a Mg$^{2+}$ transporter, has now been shown to be a subdomain of the N-linked glycosylation apparatus (Ravell et al, 2020). Moreover, no alterations in total and ionized serum Mg$^{2+}$ levels were shown in patients diagnosed with XMEN disease, who carry a loss-of-function mutation in MAGT1 (Ravell et al, 2020). The predominant interpretation seems to be that TRPM7 is majorly involved in cellular and systemic Mg$^{2+}$ homeostasis (Zou et al, 2019). Similar to many other cell types (Schmitz et al, 2003; Hardy et al, 2023; Hoeger et al, 2023; Chubanov et al, 2024), our study further supports a role for TRPM7 as the primary Mg$^{2+}$ uptake pathway in human CD4 T lymphocytes. Given that many effects of impaired TRPM7 function can be restored with Mg$^{2+}$ supplementation, also supported by the data shown here, TRPM7-independent pathways of Mg$^{2+}$ uptake must exist, for example, through transporter

TRPM7 inhibition during polarization of naïve CD4 T cells into iT$_{reg}$ cells preserves FOXP3$^+$ signals of CD25$^+$CD127$^{lo}$ iT$_{reg}$ cells. Polarization of naïve CD4 T cells into iT$_H$17 cells results in augmented RORɣt expression in the presence of 6 mM Mg$^{2+}$, which is reduced upon TRPM7 inhibition, highlighting the need for Mg$^{2+}$ uptake and related TRPM7-dependent intracellular signaling for iT$_H$17 cell polarization (right). **(B, C, E, F, G, I, J, L, M, O)** Statistics: one-way ANOVA (B, C, E, J, L) and $t$ test (F, G, I, M, O). *$P < 0.05$; **$P < 0.005$; ***$P < 0.0005$; ****$P < 0.0001$ and n.s., not significant. Data are mean ± SD.

proteins. Different potential $Mg^{2+}$ transporters, such as CNNM2 and the solute carriers SLC41A1-2, have been proposed, but findings have so far been inconclusive (Mellott et al, 2020; Bai et al, 2021).

Recently, Mendu et al. showed that mice harboring a thymus-specific deletion of TRPM7 are resistant to Concanavalin-A-induced autoimmune hepatitis (Mendu et al, 2020). In their study, Mendu et al. reported TRPM7-deleted CD4 T cells to prefer $T_{reg}$ lineage, and non-$T_{reg}$ CD4 cells to activate normally (Mendu et al, 2020). Contrary to our findings that clearly show diminished activation of human CD4 T cells after blockade of TRPM7, Mendu et al. showed that murine non-$T_{reg}$ CD4 cells can still be activated (Mendu et al, 2020). This discrepancy could be due to functional differences in human and murine cells (Mestas & Hughes, 2004). Moreover, the genetic mouse model induced altered thymocyte development and differentiation, evidenced by the majority of TRPM7-deficient murine T cells remaining in double-negative state (Faouzi et al, 2017; Beesetty et al, 2018). This is not easily comparable to physiologically differentiated human cell populations. In line with our current findings on TRPM7 affecting $Ca^{2+}$ signaling, Faouzi et al., and Beesetty et al., described TRPM7 to be linked to altered store-operated calcium entry (SOCE) in DT40 chicken B cells and in a TRPM7 kinase-deficient mouse model (Faouzi et al, 2017; Beesetty et al, 2018). Underlying key mechanisms remain unclear and demand further investigation. Importantly, previous findings support the notion that TRPM7 kinase moiety is influenced by TRPM7 channel conductance, whereas the kinase activity is not essential for channel function, but structurally important, as shown in delta-kinase models (Ryazanova et al, 2004; Romagnani et al, 2017; Nadolni et al, 2020; Hoeger et al, 2023). Since TRPM7 kinase has been shown to influence murine T-cell activation (Romagnani et al, 2017; Beesetty et al, 2018), this mechanism of connected channel and kinase function might very well cause some of the effects observed in our current study on human T cells, and will remain subject of further investigations. However, despite several available TRPM7 channel blockers, the scientific community still lacks pharmacological tools to target TRPM7 kinase specifically, posing considerable challenges to interpret the actions of TRPM7 kinase versus channel function separately.

Previously, TRPM7 kinase has been shown to impact T-cell differentiation in a kinase-deficient mouse model, as kinase-inactivated murine naïve T cells were unable to differentiate into the pathogenic $T_H17$ lineage, whereas $T_{reg}$ cell development was not impaired (Romagnani et al, 2017). Similar findings from Mendu et al. showed TRPM7-deleted CD4 T cells to prefer $T_{reg}$ lineage (Mendu et al, 2020). In line these previous studies, our results on human CD4 T cells suggest that TRPM7 influences i$T_{reg}$ and i$T_H17$ cell differentiation. This is evidenced by our observation of preserved FOXP3 and reduced RORγt expression upon TRPM7 blockade. T-cell differentiation is a multifaceted process comprising multiple cytokine signals including a lineage-specific transcription factor profile (Olson et al, 2016; Campe & Ullrich, 2021; Jiang et al, 2021). Activation of the AKT-mTOR signaling pathway can impair $T_{reg}$ cell development in vivo, whereas inhibition of this pathway (e.g., using rapamycin), combined with TCR signaling, can induce FOXP3 expression and serves as a common means to induce i$T_{reg}$ cells in vitro (Haxhinasto et al, 2008; Sauer et al, 2008; Sakaguchi et al, 2020). We here show for the first time a molecular interaction of TRPM7 with AKT1 in CD4 T cells, which is clearly reduced upon TRPM7 inhibition, in line with the observed reduction in AKT phosphorylation. Upon pharmacological inhibition of TRPM7 during the polarization of naïve CD4 T cells toward the immunosuppressive $T_{reg}$ cells, we observed a clear dose-dependent effect highlighting the importance of $Mg^{2+}$ for cellular activation and proliferation. Counterbalancing these effects, the reduced AKT activation is associated with preserved FOXP3 induction in the TRPM7-inhibited i$T_{reg}$ cells.

We recently demonstrated a direct phosphorylation of AKT and SMAD2 via the TRPM7 kinase, influencing downstream signaling in murine and human immune and leukemia cells, respectively (Romagnani et al, 2017; Nadolni et al, 2020; Hoeger et al, 2023). SMAD proteins have been reported to have diverse functions in T-cell differentiation. While SMAD2 and SMAD4 are indispensable for $T_H17$ differentiation, deletion of SMAD2 has been suggested to promote FOXP3 transcription (Martinez et al, 2010; Corral-Jara et al, 2021; Dong 2021). Of note, SMAD2 is known as a direct positive regulator of IL-17A expression (Corral-Jara et al, 2021). The reduced SMAD2 activation upon TRPM7 inhibition, as we observed during i$T_H17$ induction, may thus, in addition to inhibition of AKT signaling upon TCR stimulus, positively affect FOXP3 expression in i$T_{reg}$ cells. However, in an in vitro setting, crosstalk of pathways might be minimal. Our data indicate that TRPM7 inhibition negatively affects both SMAD2 and AKT signaling also in human CD4 T cells, resulting in reduced i$T_H17$ polarization and the induction of FOXP3 in i$T_{reg}$ cells. In addition to SMAD2, different STAT proteins have been demonstrated to be involved in T-cell differentiation (Betts et al, 2017; Zhu et al, 2021; Cheung et al, 2022; Qin et al, 2024). Cheung et al. showed the noncanonical STAT3 Ser727 phosphorylation induced via the IL-6 pathway to be important for $T_H17$ differentiation in mice (Cheung et al, 2022). We tested for both STAT3 phosphorylation sites via TCR and IL-6 stimulation, and neither of these events was influenced upon TRPM7 inhibition suggesting that TRPM7 remains primarily involved in TGF-$\beta$-SMAD2-related induction of $T_H17$ cells. This thus confirms in human T cells, what we have previously shown in murine cells (Romagnani et al, 2017). Similar to STAT3, we found no impact on STAT5 phosphorylation upon TRPM7 inhibition, suggesting that the TRPM7-mediated induction of FOXP3 was independent of STAT3 or STAT5 signaling during in vitro i$T_{reg}$ cell differentiation. Contrarily to our findings, Mendu et al (2020) reported an increase in pSTAT5 signals in T cells isolated from a Lck-Cre transgenic mouse model where *Trpm7* was deleted. The *Lck* gene is primarily expressed in T cells of the thymus, playing an essential role in selection and maturation of developing T cells, overall leading to a developmental block of thymocytes by keeping them in a double-negative state (Jin et al, 2008; Mendu et al, 2020). This might not only account for compensatory effects, but in addition makes it difficult to compare TRPM7 (in-)dependent signaling effects of these cells with the pharmacological approach in fully developed human CD4 T cells. Additionally, functional differences have been reported for the murine and human system (Mestas & Hughes, 2004). SMAD2 has also been shown to act as a co-activator of STAT3, both counteracting STAT5 binding and further driving the pro-inflammatory $T_H17$ lineage (Yoon et al, 2015; Corral-Jara et al, 2021). Similar to STAT5, FOXP3 was shown to inhibit RORγt mediated $T_H17$ differentiation (Hirahara et al, 2010; Qin et al,

2024). The enhanced FOXP3 expression upon TRPM7 inhibition may play an additional role in the inhibition of $iT_H17$ cells, in addition to dampened SMAD2 signaling. Collectively, in conjunction with murine data shown by Mendu et al (2020) and Romagnani et al (2017), our data on human CD4 T cells suggest TRPM7-mediated $T_H17$ differentiation to be driven via SMAD2, having no or little impact on STAT3/5 signaling. Notably, here we demonstrate for the first time that the impact of TRPM7 on pro-and anti-inflammatory T-cell homeostasis may be translated from mouse to human.

In summary, TRPM7 is an important regulator of human T-lymphocyte function stirring not only immune system homeostasis, but potentially also lymphatic malignancies. As a crucial pathway for $Mg^{2+}$ entry, TRPM7 regulates T-cell signaling by influencing $Ca^{2+}$ and $Mg^{2+}$-dependent cellular activation processes. While further research into TRPM7 and its effects on immune cell function including TRPM7 kinase-related signaling is needed, this study underlines TRPM7 as a possible druggable target in T-cell-dependent pro-inflammatory and autoimmune diseases.

# Materials and Methods

### Jurkat T cells and cell culture

TRPM7-deficient (clone KO1, E12, and clone KO2, A03, both Thermo Fisher Scientific) Jurkat T-cell clones were generated by CRISPR-Cas9 genome editing at Thermo Fisher Scientific (US). sgRNA was designed to target *TRPM7* in exon 4 (target sequence: 5'TGATCC ATAAGCATCCGTT-3') and using *TRPM7*-directed ss-oligos (5' GEZGACCATTTTAATCAGGCAATAGAAGAATGGTCTGTGGAAAAGCATA-CAGAACAGAGCCCATAGGATGCTTATGGAGTCATAAATTTTCAAGGGGEZT 3'), creating a single base pair insertion resulting in a premature stop codon. In short, E6-1 Jurkat T cells were co-transfected with Cas9, ss-oligos and in vitro-transcribed (IVT) gRNA. Stable pools were undergone NGS analysis, and stable clones were generated by limiting dilution expansion. Two KO clones were retrieved and confirmed by Sanger sequencing, compared with untargeted WT cells. Primary lymphocytes and Jurkat T cells (Jurkat E6.1 [WT]) were cultured in Roswell Park Memorial Institute (RPMI) medium containing 10% heat-inactivated FBS and 1% penicillin/streptomycin in a humidified atmosphere at 37°C containing 5% $CO_2$. Medium of KO cells was supplemented with 6 mM $MgCl_2$. $MgCl_2$ supplement was removed 21 h before measurements.

### Primary human T-cell isolation

Cells were isolated from peripheral blood of healthy donors according to the respective ethics approvals. PBMCs were isolated by density gradient centrifugation using Lymphoprep (Stemcell Technologies). Isolation of respective lymphocyte subsets was achieved using magnetic cell specific separation kits. For naïve CD4 T cells, EasySep Human Naïve CD4 T Cell Isolation Kit II was used. For total CD4 T cells, the EasySep Human CD4T Cell Isolation Kit was used. For both $CD4^+$ $CD25^-$ effector cells and $CD4^+$ $CD25^+$ $T_{reg}$ cells, EasySep Human $CD4^+CD127^{low}CD25^+$

Regulatory T Cell Isolation Kit was used, according to the manual. A minimum of two different donors were used in primary human T-cell experiments.

### TRPM7 inhibitors

Synthetic TRPM7 inhibitor NS8395 was purchased from Alomone.

Waixenicin A as a natural compound TRPM7 inhibitor was isolated as following: freeze-drive biomass of Sarcothelia edmonsoni Verill, 1928 was ground and extracted with hexane. After removal of solvent and elution through a C18 solid phase extraction column, the extract was subjected to reversed phase HPLC (column: Sili-Cycle dt C18, 30 × 100 mm, 5 $\mu$m; mobile phase: acetonitrile/water gradient, 50–80% acetonitrile from 0–2 min, 80–100% acetonitrile from 2–6 min; 100% acetonitrile from 6–12 min). Waixenicin A eluted at 6.01 min and was aliquoted into 50 $\mu$g single use vials. Purity was confirmed at >95% by LC-MS with evaporative-light scattering detector.

### Electrophysiology

TRPM7 currents were acquired via whole-cell patch clamp. A ramp from −100 to +100 mV over 50 ms acquired at 0.5 Hz and a holding potential of 0 mV was applied. Inward and outward current amplitudes were extracted at −80 and +80 mV, respectively. Data were normalized to the cell size measured after whole-cell break-in (pA/pF). Capacitance was measured using the capacitance cancellation (EPC-10; HEKA). $Mg^{2+}$-free extracellular solution (in mM): 140 NaCl, 3 $CaCl_2$, 2.8 KCl, 10 HEPES-NaOH, 11 glucose (pH 7.2, 290–300 mOsm/liter). Intracellular solution (in mM): 120 Cs-glutamate, 8 NaCl, 10 Cs-EGTA, 5 EDTA (pH 7.2, 290–300 mOsm/liter).

### Proliferation and viability measurements

Jurkat T cells were seeded at a density of 500,000 cells into 24-well plates and cultured in normal RPMI or RPMI with 6 mM $MgCl_2$ for 5 d. Proliferation was analyzed daily using Guava ViaCount reagent on a Guava Easycyte 12HT flow cytometer (Cytek Bioscoences). Proliferation experiments on primary T cells followed a similar procedure. Alternatively, T cells were stained with CFSE dye (1 $\mu$M; Biozym), washed, and cultured for 5 d, before monitoring proliferation traces (dye dilutions) on a BC Cytoflex flow cytometer.

### Inductively coupled plasma mass spectrometry

$Mg^{2+}$ content was determined by inductive couple plasma mass spectrometry (ICP-MS) by ALS Scandinavia (Sweden). Jurkat WT and KO cells were incubated overnight in RPMI ± 6 mM $MgCl_2$, washed 2x with dPBS (w/o $Mg^{2+}$ or $Ca^{2+}$; Sigma-Aldrich). Likewise, Jurkat WT cells were cultured overnight in RPMI ± 6 mM $MgCl_2$ containing 30 $\mu$M NS8593. Cells were seeded with a density of 5 × $10^6$ cells per condition, cell pellets were dried overnight at 70°C and stored at −80°C. Collected samples were shipped on dry ice for further analysis via ICP-MS.

## Jurkat T-cell Ca²⁺ imaging

Jurkat T cells were loaded with 3 $\mu$M Fura-2 AM and 0.05% PluronicF-127 (Invitrogen) in imaging buffer, 15 min at 37°C. Cells were washed with imaging buffer to remove excess dye. Imaging buffer consisted of $Ca^{2+}$- and $Mg^{2+}$-free HBSS supplemented with (in mM): 2 $CaCl_2$, 0.4 $MgCl_2$, 10 glucose. Cells were seeded into Poly-D-lysine precoated $\mu$-Slide eight-well high, chambered coverslips (IBIDIs) and incubated for 10 min before start of the measurement. Time lapse images were acquired on an AnglerFish imaging system (Next Generation Fluorescence Imaging/NGFI), using 5 $\mu$M thapsigargin (Thermo Fisher Scientific) to mobilize $Ca^{2+}$ from intracellular stores. The TRPM7 channel inhibitor NS8593 was used at a concentration of 30 $\mu$M. Viable cells, identified by their ionomycin response at the end of the measurement, were analyzed with Fiji.

## Ca²⁺ imaging of primary T cells

Primary CD4 T cells were loaded with 3 $\mu$M Fura-2 AM in RPMI supplemented with 10% FBS, 30 min at 37°C while in reaction tubes. Cells were washed twice with imaging buffer to remove excess dye. Imaging buffer contained (in mM): 140 NaCl, 2 $CaCl_2$, 1 $MgCl_2$, 2.8 KCl, 10 HEPES-NaOH, 11 glucose (pH 7.2, 290–300 mOsm/liter). Cells were incubated for 15 min at RT and then slowly pipetted onto chambered, antibody-coated ($\alpha$-CD3 [clone OKT3]/$\alpha$-CD28 [clone CD 28.2], both Thermo Fisher Scientific) coverslips on focus plane. Intracellular $Ca^{2+}$ was monitored with Fura-2 AM (SantaCruz) using dual excitation at 340 and 380 nm, detection at 520 nm. Fluorescence images were acquired on a TillVisIon imaging system (TILL photonics). Quantification of basal (before stimulation), delta $Ca^{2+}$ (difference between maximum $Ca^{2+}$ influx and basal levels), AUC (time interval 10–800 s), and oscillation frequency (number of $Ca^{2+}$ oscillations/time) was calculated.

## Immunofluorescence staining

Localization of NFATc1 was acquired on a Zeiss LSM 780 microscope or Zeiss LSM 900 confocal microscope, using a 63x oil objective. Jurkat T cells were stimulated with 5 $\mu$M thapsigargin for 30 min or left unstimulated. Primary human T cells were stimulated with plate-bound $\alpha$-CD3/$\alpha$-CD28 antibodies for 45 min. TRPM7 channels were inhibited using 30 $\mu$M NS8593 and compared against cells treated with DMSO as solvent control. Cells were permeabilized with 0.1% Triton X-100 for 5 min and stained for intracellular NFAT using $\alpha$-NFATc1 antibody (1:100, #7A6; Santa Cruz) in 0.2% BSA/1% normal goat serum in PBS, and secondary $\alpha$-mouse antibody AF647 (1:1,000; Cell Signaling). Cells were counterstained with DAPI (0.2 $\mu$g/ml) and mounted onto glass coverslips using Antifade ROTI-Mount FluorCare (Carl Roth). Zen 3.5 software was applied. Nuclear NFAT levels were analyzed; therefore, regions of interest (ROI) were defined by nuclear outlines (DAPI signals). AF647 signal intensity was corrected by background signals.

## Flow cytometry of activation markers

Lymphocytes were seeded in 96-well plates at 2 × 10⁵ cells per condition in 100 $\mu$l RPMI with 10% FBS. Cells were treated with 0.1%

DMSO, NS8593 (30, 20 or 10 $\mu$M, as indicated), or 6 mM $MgCl_2$ as indicated. 15 min after treatment, cells were stimulated with antibodies against CD3/CD28 (2 $\mu$g/ml $\alpha$-CD3 and 1 $\mu$g/ml $\alpha$-CD28 antibodies, ImmunoCult Human $\alpha$-CD3/$\alpha$-CD28 T Cell Activator, Stemcell Technologies, or eBioscience) or PMA (20 ng/ml) and ionomycin (1 $\mu$g/ml) (both from Sigma-Aldrich). After 24 or 48 h, respectively, cells were stained according to the manufacturer's instructions. Cells were washed twice after staining. Isotype controls or FMO controls were performed. Cells were analyzed using a Guava Easycyte 6-2L flow cytometer (Luminex Corporation), or a Beckman Coulter CytoFLEX. The following antibodies were used: $\alpha$-human CD4-VioBlue (REA623; Miltenyi), $\alpha$-human CD45RA-APC-Vio770 (REA562; Miltenyi), $\alpha$-human CD69-APC (REA824; Miltenyi), $\alpha$-human CD25-VioBright515 (REA570; Miltenyi).

## IL-2 quantification

Lymphocytes were seeded in 96-well plates at 2 × 10⁵ cells per conditions in 100 $\mu$l RPMI with 10% FBS. Cells were treated with 0.1% DMSO, 30 $\mu$M NS8593, or 6 mM $MgCl_2$ as indicated. 15 min after treatment, cells were stimulated with antibodies against CD3/CD28 (ImmunoCult Human $\alpha$-CD3/$\alpha$-CD28 T Cell Activator, Stemcell Technologies, as before). Cell supernatants were collected 48 h after cell stimulation and stored at –80°C. IL-2 concentrations were analyzed using a Biogems Precoated Human IL-2 ELISA kit (Biogems International, Inc.) according to manufacturer's instructions by measuring absorbance at 405 nm on a BMG Labtech Clariostar Plus plate reader.

## mRNA isolation

Jurkat TRPM7 KO cells were cultured overnight in normal RPMI without additional $MgCl_2$ supplementation, KO cells, and WT cells were seeded at a density of 4 × 10⁶ cells per condition and stimulated for 3 h with 10 ng/$\mu$l PHA. mRNA was isolated from cell pellets using RNeasy Mini Kit (QIAGEN) following manufacturer's instructions. mRNA concentrations were determined via OD measurement.

## cDNA synthesis and quantitative real-time PCR (qRT-PCR)

For cDNA synthesis, 0.5 $\mu$g mRNA was diluted in $H_2O$, mixed with 0.05 ng random hexamers and 0.5 $\mu$g Oligo(dT)$_{15}$ (Promega) and incubated for 5 min at 70°C. On ice, 1 mM dNTPs, 5x buffer, M-MLV Reverse Transcriptase (all from Promega), and DEPC-treated $H_2O$ (Sigma-Aldrich) were added and incubated for 60 min at 42°C. The resulting cDNA was diluted 1:4. Transcripts were analyzed by specific primer pairs. Master mixes additionally contained cDNA and SYBR-GreenTM (Sigma-Aldrich). Transcripts were measured in technical triplicates on a CFX-96 cycler (Bio-Rad): 50°C 2′, 95°C 10′ (preincubation), 95°C 15″, 62°C 30″, 72°C 30″, 40 cycles (amplification), 95°C 10″, 60°C 1′ (melting), 40°C 10′ (cooling). Primer pairs (all human, 5′-3′), h*IL-2* (fw) TTTACATGCCCAAGAAGGCC and (rev) GTT GTTTCAGATCCCTTTAGTTCCA, h*TRPM7* (fw-1) GTCAGCAACTCGTCGGTG TT and (rev-1) GATAGCCTCACTACTTAGCTCTGTAGGA, h*TRPM7* (rev-2) TTGGTGTCATATGATAGCCTCACATA, were used in combination with h*TRPM7* fw-1, h*TRPM7* (fw-2) ATCAGGCAATAGAAGAATGGTCTGT in

combination with h*TRPM7* (rev-3) CATGTTTTGCCACACCTGTGT and h*HPRT1* (fw) CCCTGGCGTCGTGATTAGTG, and (rev) TCGAGCAAGACG TTCAGTCC. For analysis, data on *TRPM7* primer pairs were pooled. A minimum of three independent experiments were performed. CT values of *HPRT* transcripts were subtracted from measured transcript CT values, to calculate $2^{(-\Delta CT)}$.

### iT$_{reg}$ and iT$_H$17 cell differentiation and flow cytometry

Naïve CD4 T cells were seeded at a density of $1 \times 10^5$ cells per condition into a 96-well plate and treated with 10–30 $\mu$M NS8593, equivalent volume of DMSO, or 6 mM MgCl$_2$, equivalent volume of H$_2$O, or 2.5–10 $\mu$M Waixenicin A or equivalent volume of EtOH. iT$_{reg}$ cell induction medium contained $\alpha$-CD3/$\alpha$-CD28 dynabeads (Thermo Fisher Scientific), 10 ng/$\mu$l rhIL-2 (Immunotools), 5 ng/$\mu$l TGF-ß (Immunotools), and 100 nM ATRA (Sigma-Aldrich). Cells were cultured for 6 d in a humidified atmosphere at 37°C containing 5% CO$_2$, with intermediary medium exchange on day 4 and transfer to a 24-well plate. Surface staining was performed using the following antibodies: $\alpha$-human CD4-VioBlue (REA623; Miltenyi), $\alpha$-human CD25-PE (BC96; BioLegend), $\alpha$-human CD45RA-APC-Vio770 (REA562; Miltenyi), $\alpha$-human CD127-PE-Vio615 (REA614; Milteny). Surface stain included the Viobility 405/452 Fixable Dye (Milteny), followed by fixation and permeabilization using the "Fix/Perm" buffer (Thermo Fisher Scientific). Signature transcription factor was stained using the following antibody: $\alpha$-human FoxP3-PE (REA1253; Miltenyi). Cells were analyzed using a Cytek Northern Lights 3,000, applying spectral unmixing for antibody- and drug-based effects. Data were analyzed using FlowJo v10.9 or higher. Naïve CD4 T cells were used as gating control.

Naïve CD4 T cells were seeded at a density of $1 \times 10^5$ cells per condition into a 96-well plate and treated with 10–30 $\mu$M NS8593, equivalent volume of DMSO or 6 mM MgCl$_2$ or equivalent volume of H$_2$O. iT$_H$17 induction medium contained 30 ng/ml IL-6, 100 ng/ml IL-21, 30 ng/ml IL-23, 5 ng/ml TGF-ß, 10 $\mu$g/ml a-INF$\gamma$, and 10 $\mu$g/ml a-IL-4. Cells were cultured for 6 d. On day 6, restimulation was performed for 4 h with 20 ng/ml PMA (Sigma-Aldrich) and 1 $\mu$g/ml ionomycin, including 5 $\mu$g/ml Brefeldin A (Thermo Fisher Scientific). Surface staining was performed using the following antibodies: $\alpha$-human CD25-PE (BC96; BioLegend), $\alpha$-human CCR6-PEVio770 (REA190; Milteny). Surface stain contained the Viobility 405/452 Fixable Dye, followed by fixation with 2% formaldehyde in PBS and permeabilization (0.1% Saponin, 1% FBS, 0.005% NaN$_3$ in PBS). Signature transcription factor was stained using the following antibody: $\alpha$-human ROR$\gamma$t-PE (REA278; Milteny) and $\alpha$-human FOXP3-APC (REA1253; Miltenyi). Cells were analyzed using a Cytek Northern Lights 3,000, applying spectral unmixing for antibody- and drug-based effects. Data were analyzed using FlowJo v10.9 or higher. Naïve CD4 T cells were used as gating control.

### SDS–PAGE and Western blot

CD4 T cells were seeded at a density of $1 \times 10^6$ cells per condition and stimulated with 5 ng/ml TGF-$\beta$ (Immunotools or Peprotech) or 30 ng IL-6 (Immunotools) or 10 ng IL-2 (Immunotools) or $\alpha$-CD3/$\alpha$-CD28 (Thermo Fisher Scientific) for the indicated time frame. Lysates were prepared in RIPA (25 mM Tris–HCl pH7.5, 150 mM NaCl,

1 mM EDTA, 1% NP-40, 1% C$_{24}$H$_{39}$NaO$_4$, 0.1% SDS) and diluted with 4x Laemmli buffer (62.5 mM Tris/HCl, 20% [vol/vol] glycerol, 5% [vol/vol] $\beta$-mercaptoethanol, 4% [wt/vol] SDS, 0.1% [wt/vol] bromophenol blue), heated to 95°C for 10 min and subjected to SDS–PAGE. Proteins were transferred to a polyvinylidene fluoride membranes (Thermo Fisher Scientific) by Western blotting and blocked with 5% BSA or skim milk in TBST buffer. Membranes were incubated in respective antibodies according to standard procedures. The following antibodies were used: $\alpha$-pSMAD2 Ser465/ Ser467 (138D4; Cell Signaling), $\alpha$-SMAD2/3 (D7G7; Cell Signaling), $\alpha$-AKT1 (D9-9-C9; Thermo Fisher Scientific), $\alpha$-pSTAT3 Tyr705 (D3A7; Cell Signaling), $\alpha$-pSTAT3 Ser727 (D8C2Z; Cell Signaling), $\alpha$-pSTAT5 Tyr 694 (D47E7; Cell Signaling), $\alpha$-GAPDH (G-9; Santa Cruz). Secondary antibodies: IgG (H+L) goat $\alpha$-rabbit, HRP-conjugated (15217664; Thermo Fisher Scientific), $\alpha$-mouse IgG, HRP-linked (7076P2; Cell Signaling). Immune reactivity was quantified via densitometry (Bio-Rad). Samples were normalized to respective loading controls and experimental controls. Data were analyzed with Fiji.

Western blot lysates to stain for TRPM7 were prepared as following: Jurkat TRPM7 WT and KO cells were seeded at a density of $2.5 \times 10^6$ cells/condition. Lysates were prepared in Pierce IP Lysis Buffer (Thermo Fisher Scientific) and diluted with 4x Laemmli buffer, heated to 65°C for 10 min and subjected to SDS–PAGE. Proteins were loaded onto a 7.5% SDS gel, transferred to a nitrocellulose membrane (Amersham Protran) by Western blotting and blocked with 5% BSA in TBST buffer. Membranes were incubated with the respective antibodies according to standard procedures. The following antibodies were used: $\alpha$-TRPM7 (ACC-047; Alomone) or $\alpha$-HSP90 (C45G5; Cell Signaling).

### Intracellular cytokine and transcription factor staining

CD4 T cells were stimulated as for Western blot analysis. Upon stimulation, cells were fixed in BD Cytofix buffer (BD Biosciences), following permeabilization using BD Perm Buffer III (BD Biosciences) and subsequent staining. Staining was performed using the following antibodies: $\alpha$-human pSMAD2 PE Ser465/S467 (E8F3R; Cell Signaling), $\alpha$-human pSTAT3 APC Tyr705 (BD), $\alpha$-human pSTAT3 FITC Ser727 (BD), and $\alpha$-human pSTAT5 APC Tyr694 (BD). Cells were analyzed using a Guava Easycyte 6-2L flow cytometer (Luminex Corporation), and data were analyzed using FlowJo v10.9 or higher.

### Proximity-ligation assay

Proximity ligation was performed using the Duolink in situ PLA detection kit (cat#: DUO92101; Sigma-Aldrich). CD4 T cells were seeded at a density of $1 \times 10^6$ cells per condition and stimulated for 30 min with antibodies against CD3/CD28 (Thermo Fisher Scientific) for AKT1-TRPM7 interactions, or for 10 min with 5 ng/ml TGF-$\beta$ (Peprotech) to detect SMAD2-TRPM7 interaction, and subsequently fixed with 2% formaldehyde in for 9 min at RT and permeabilized with 0.1% Triton X-100 in PBS for 5 min at RT. Blocking and ligation procedure were done according to the kit manual. Cells were stained with primary antibodies overnight at 4°C, $\alpha$TRPM7 (ACC-047; Alomone or MA527620; Thermo Fisher Scientific) or $\alpha$-AKT1 (D9-

9-C9; Thermo Fisher Scientific) or $\alpha$-SMAD2 (sc-101153; Santa Cruz). Slides were detected using a LSM900 confocal microscope with a 63X oil objective by selecting a minimum of three random selected fields. Images were analyzed via Fiji. Regions of interest (ROI) were defined by cell membranes in the brightfield image. PLA signals were counted per cell; therefore, images were background corrected by setting a lower threshold. PLA signals were counted using the Fiji Plug In "Analyze Particles" selecting for a particle size of 0.1–1 mycon$^2$. Particle count per cell was used for statistical comparison.

### Statistics, data presentation, and schematic illustration

Data were plotted using Graphpad Prism 8 (Graphpad Software) or higher. Statistical analysis of the difference of two data sets was performed using $t$ test or Mann-Whitney $U$ test. Comparison of three or more data sets was performed using one- or two-way-ANOVA, Kruskal-Wallis test, or Friedmann test, depending on the respective experimental design. Graphical illustrations were created with BioRender.

### Study approval

In this study, healthy volunteers of both sexes were enrolled. Informed consent was obtained from all participants. Peripheral blood of healthy volunteers was obtained by venipuncture. The study was conducted according to the guidelines of the Declaration of Helsinki and, approved by the local ethics boards of the Johannes Kepler University Linz (EK 1064/2022) as well as the Ludwig-Maximilians-Universität München (Az.21-1288).

# Data and Material Availability

Materials may be requested from the corresponding author, upon reasonable request.

# Supplementary Information

# Acknowledgements

We thank Viktoria Sperrer for her excellent technical assistance. Authors thank the following funding agencies: K Hoelting was supported by the FoeFoLe program (LMU Munich); B Karner-Hoeger thanks the Johannes Kepler University Impetus program (I-18-23). FD Horgen received support from NIH NIGMS P20GM103466. A Dietrich, T Gudermann, and S Zierler were supported by the Deutsche Forschungsgemeinschaft (DFG, German Research Foundation) TRR 152/1-3 Project 14 (S Zierler), 15 (T Gudermann) and 16 (A Dietrich). The usage of devices from the JKU Core Facilities Flow Cytometry and Imaging is highly acknowledged. Supported by Johannes Kepler University Open Access Publishing Fund and the Federal State Upper Austria.

## Author Contributions

A Madlmayr: conceptualization, formal analysis, investigation, visualization, project administration, and writing—original draft, review, and editing.
K Hoelting: conceptualization, formal analysis, funding acquisition, investigation, and writing—original draft.
B Karner-Hoeger: formal analysis, funding acquisition, investigation, and writing—original draft, review, and editing.
D Lewitz: investigation.
M Weng: investigation.
S Hacker: investigation.
J Eder: investigation.
K Horner: investigation.
C Schedlberger: investigation.
T Haider: investigation.
M Lechner: investigation.
M Duggan: investigation.
R Ross: investigation.
FD Horgen: resources, supervision, funding acquisition, and writing—review and editing.
M Sperandio: resources and writing—review and editing.
A Dietrich: resources, supervision, funding acquisition, and writing—review and editing.
T Gudermann: resources, supervision, and funding acquisition.
S Zierler: conceptualization, resources, supervision, funding acquisition, project administration, and writing—review and editing.

## Conflict of Interest Statement

The authors declare that they have no conflict of interest.

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
