## [Reviewer comments · Life Science Alliance]

TRPM7 and magnesium orchestrate human CD4 T-cell activation and differentiation

Anna Madlmayr, Kilian Hoelting, Birgit Karner-Hoeger, Dorothea Lewitz, Marius Weng, Severin Hacker, Julia Eder, Katharina Horner, Christine Schedlberger, Tanja Haider, Max Lechner, Michelle Duggan, Rylee Ross, F. David Horgen, Markus Sperandio, Alexander Dietrich, Thomas Gudermann, and Susanna Zierler

DOI: <https://doi.org/10.26508/lsa.202503357>

Corresponding author(s): Susanna Zierler, Johannes Kepler University of Linz

Review Timeline:

Submission Date:	2025-04-11
Editorial Decision:	2025-06-02
Revision Received:	2025-10-10
Editorial Decision:	2025-11-03
Revision Received:	2025-11-10
Accepted:	2025-11-16

Scientific Editor: Tim Fessenden

Transaction Report:

June 2, 2025

Re: Life Science Alliance manuscript #LSA-2025-03357-T

Prof. Susanna Zierler
Johannes Kepler University Linz
Institute of Pharmacology
MED Campus I, Krankenhausstr. 5
Linz 4020
AUSTRIA

Dear Dr. Zierler,

Thank you for submitting your manuscript entitled "TRPM7 and magnesium orchestrate human CD4 T-cell activation and differentiation" to Life Science Alliance. The manuscript was assessed by expert reviewers, whose comments are appended to this letter.

As you will see, reviewers were uniformly positive about the advance set forth in this work on cation handling and T cell differentiation and function related to TRPM7. Reviewers 1 and 2 expressed concerns mostly related to technical aspects of this work. Namely, both of these reviewers sought improved western blots and flow staining of ROR γ T levels. Both also requested validation of NS8593 effects on Jurkat T cells. Next both Reviewers 2 and 3 requested validation of the KO approach, and here we concur that the sgRNA sequence must be provided as well as confirmation of KO efficacy. Reviewer 1 noted the causal link between Mg $^{2+}$ handling by TRPM7 and Ca $^{2+}$ signaling, which agree is central to this work and should be more thoroughly discussed in the text. Finally Reviewer 2 requested reanalysis of Ca $^{2+}$ plots. While all reviewer comments should be addressed in some form, additional data beyond those mentioned here are not required in a revised manuscript.

While you are revising your manuscript, please also attend to the below editorial points to help expedite the publication of your manuscript. Please direct any editorial questions to the journal office. The typical timeframe for revisions is three months. Please note that papers are generally considered through only one revision cycle, so strong support from the referees on the revised version is needed for acceptance.

We hope that the comments below will prove constructive as your work progresses. Thank you for this interesting contribution to Life Science Alliance. We are looking forward to receiving your revised manuscript.

Sincerely,

-- Summary blurb (enter in submission system): A short text summarizing in a single sentence the study (max. 200 characters including spaces). This text is used in conjunction with the titles of papers, hence should be informative and complementary to

the title and running title. It should describe the context and significance of the findings for a general readership; it should be written in the present tense and refer to the work in the third person. Author names should not be mentioned.

B. MANUSCRIPT ORGANIZATION AND FORMATTING:

Reviewer #1 (Comments to the Authors (Required)):

In this study, Madlmayr et al. investigate how TRPM7 affects the function of human T cells using CRISPR mediated deletion of TRPM7 (Jurkat T cell line) or a TRPM7 inhibitor (primary CD4 T cells). They show that deletion of TRPM7 in Jurkat cells inhibits TRPM7 currents, calcium influx, NFAT translocation and cell proliferation and viability. Similar results are obtained in primary CD4 T cells treated with the TRPM7 inhibitor NS8593. Intriguingly, anti-CD3 induced calcium influx is not abolished by NS8593, but calcium oscillations are. The authors link suppressed calcium oscillations to reduced production of IL-2 and expression of CD25 and CD69 by CD4 T cells. Mechanistically the most interesting part of the study are the data in figure 7, where the authors describe that NS8593 inhibits the phosphorylation of AKT and SMAD2, and blocks interaction TRPM7 with SMAD2. Moreover, NS8593 enhances the expression of Foxp3 but reduces that of RORgt during CD4 T cell differentiation in vitro, suggesting that TRPM7 regulates the balance of iTreg and Th17 cell fates. Overall the study is very interesting, well executed, the data are mostly convincing and the manuscript is well written. The results are consistent with the senior author's previous results in mouse T cells using mice expressing a kinase dead form of TRPM7, suggesting that the role of TRPM7 in mouse and human T cells is conserved. Most of my concerns are technical in nature, with one conceptual question regarding the role of Mg²⁺ vs Ca²⁺ in TRPM7 and T cell function.

Major comments:

1. Using ICP/MS to measure total cellular Mg²⁺ in TRPM7 deficient Jurkat cells or Jurkat cells treated with NS8593, the authors show that Mg²⁺ levels are reduced in the absence of TRPM7. This is in contrast to the original TRPM7 knockout paper in T cells by Jin et al. 2008 (Ref 13; minor comment: please fix format of Ref.13). How do the authors explain this discrepancy? And does this reduction of total cellular Mg²⁺ matter for T cell function? I am asking because most subsequent analyses in this manuscript focus on the role of Ca²⁺ signaling (and not Mg²⁺) in TRPM7 deficient or inhibited T cells. This is also an important question because the authors measure Ca²⁺ flux in TRPM7 deficient T cells and relate that finding to impaired T cell function, but they then add MgCl₂ to TRPM7 cells to rescue their function. This raises the question whether reduced Ca²⁺ in TRPM7 deficient T cells is an epiphenomenon. (Another question is how does MgCl₂ enter TRPM7 deficient T cells).
2. The fact that TRPM7 deletion does not affect CD69 expression in Jurkat cells whereas treatment with NS8593 does reduce CD69 levels raises concerns about the specificity of NS8593 for TRPM7 inhibition. The authors somewhat mitigate these concerns by showing that a potential off-target of NS8593, the potassium channel SK2, does not affect T cell function when inhibited with apamin. But there may be many other off-targets of NS8593 besides SK2. A better experiment in my opinion, which is easy to do, would be to show that NS8593 does not further suppress the function of TRPM7 deficient Jurkat cells (CD69, proliferation etc).
3. I am not entirely convinced of the pAKT and pSMAD2 data in Figure 5C-I. The housekeeping control in C (GAPDH) shows great variations, and it is unclear how pAKT was normalized in D. The best normalization would be to total AKT levels. Similar concerns for pSMAD2 in E. The flow cytometry data for pSMAD2 in I show reduced phosphorylation, but that is not really apparent in the histogram in H.
4. Along similar lines, I am not fully convinced that MgCl₂ and NS8593 affect RORgt levels in T cells. The flow data in Figure 5P do not show a robust difference and it is unclear how data were quantified in Q. Please also show flow data for RORgt expression in cell treated with NS8593 (panel R).

Minor comments:

1. Mention in Figure 1A,F or its legend that TRPM7 currents were extracted at both +80 and -80 mV.
2. In Figure 2C and I, please show NFATc1 staining without DAPI. It is otherwise difficult to see NFATc1 location. For the quantification of NFATc1 location, it would be better to show ratios of nuclear to cytoplasmic NFATc1 per cell rather than intensities. Ditto in Figure 3M-P. The authors could add line scan traces as in 3M and O to Figure 2 to better visualize nuclear translocation. In Figure 3M, the IF images do not show a discernable difference in NFATc1 location, whereas the line scan data do. Are these data derived from the same cell?
3. In Figure 3C, D, I and J, what does "delta Ca²⁺" mean?
4. For the data in Figures 3-5, the authors compare effects of TRPM7 inhibition on naive CD4 T cells and conventional CD4 T cells. There are no data in the manuscript describing the properties of these subsets. Which markers do these cells express, and have the authors analyzed these post isolation?
5. In Figure 5B, I suggest to add statistical significance between cells treated with the same concentration of NS8593 in the presence or absence of MgCl₂.

Reviewer #2 (Comments to the Authors (Required)):

In the study titled "TRPM7 and magnesium orchestrate human CD4 T-cell activation and differentiation," Madlmayr et al. investigate the role of the channel-kinase TRPM7 in regulating Mg²⁺ homeostasis during the activation, differentiation, and function of human CD4 T cells. They utilize both pharmacological inhibition and genetic ablation of the protein. The authors primarily focus on human CD4 T cells, employing a widely accepted T cell leukemia cell line, Jurkats, as well as primary CD4⁺ T cells isolated from healthy donors, which broadens the scope of their research. The study is well-designed, and the methodologies used are appropriate. However, I have a few recommendations that could help the authors enhance the manuscript and present more accurate information. Overall, I believe the study is relevant, as it represents a significant advancement in understanding the role of TRPM7 in human immune cells, particularly since the mechanisms of Mg²⁺ transport and homeostasis are still debated. I encourage the authors to address these major points, as doing so could expand the scope of their manuscript and improve its suitability for publication in Life Science Alliance. Please find my comments outlined below:

Major points:

1. The sequence of the sgRNA used to target TRPM7 for generating Jurkat clones 1 and 2 must be provided, at least in the methods section. A Western blot indicating the KO of TRPM7 would be ideal; however, if this is not possible due to the lack of good antibodies, the authors should either provide the sgRNA sequence or the indel introduced by the sgRNA (sequencing results). Currently, the only evidence for the KO of TRPM7 in Jurkat cells is the absence of TRPM7 currents observed through patch clamp electrophysiology. While this method is acceptable, it is important to note that the authors used different concentrations of Mg²⁺ (i.e., 6 mM) to maintain the TRPM7-KO Jurkats alive, and Mg²⁺ is known to inhibit TRPM7. An alternative way to demonstrate the KO of TRPM7 in Jurkats would be to challenge the cells with Naltriben, a proven antagonist of TRPM7 that potentiates TRPM7 function independent of Mg²⁺ concentration (PMID: 24633576, PMID: 37156763).
2. In Figures 2B, 2H, and Supplementary Figure 1G, the authors should subtract the baseline when calculating the area under the curve (AUC) of intracellular calcium concentration. Generally, increases in intracellular Ca²⁺ concentrations are nonlinear, which can lead to an overestimation of low intracellular concentrations. A double-log transformation is needed to linearize these values. By quantifying the changes using AUC without subtracting the baseline, the changes may appear small due to the higher baseline compared to the delta. Subtracting the baseline will provide a more accurate reflection of the differences in change.
3. Are the calcium oscillations shown in Figures 3B, 3H, and Supplementary Figure 2F representative of a single cell or an average? It is difficult to visualize calcium oscillations from averaged cells, as they are not synchronized. Please clarify this more accurately in the figure legend.
4. In lines 129-140, the authors discuss their results regarding CD69 expression with the genetic deletion of TRPM7 and pharmacologic inhibition of the channel, in light of their calcium-NFAT data. To my knowledge, CD69 and other activation markers in T cells, such as CD44, are calcium-NFAT independent. Specifically, CD69 is dependent on NF-κB and AP-1, and therefore its expression should not be significantly affected by reduced calcium signaling. TCR stimulation activates multiple signaling pathways, including AP-1, NF-κB, and calcium-NFAT, which may synergize; however, reduced calcium signaling shouldn't have a strong influence on CD69 expression. IL-2 and IL-2Ra are both fully dependent on NFAT, making them good readouts for a lack of calcium-NFAT signaling in T cells. The fact that the inhibitor NS8593 shows a stronger inhibition of CD69 in WT cells may suggest that the inhibitor acts on an additional target other than TRPM7. One way to test this discrepancy would

be to apply NS8593 to the TRPM7-KO Jurkats and analyze CD69 expression to confirm that the effect is on-target (in other words, no reduction in CD69 would be expected).

5. How is the quantification of the Western blot in Figure 4C conducted? Are the authors normalizing to GAPDH or to the control at the 0-time point? GAPDH is not the best housekeeping for normalizing transcriptional and protein levels in T cells due to the strong glycolytic switch that occurs upon TCR stimulation. Although the time points used in this study are short and should not significantly affect overall GAPDH expression, it is important to note that activation can trigger a changes in GAPDH levels. Normalizing phospho-AKT (Ser473) to total AKT would be a better approach.

6. The authors use the terms "naïve CD4 T cells" and "conventional CD4 T cells" to distinguish their cells based on the CD45RA expression. In principle, all naïve T cells are conventional T cells as they have not differentiated towards the regulatory subset. I will be more cautious with that terminology and simply use naïve CD4 T cells (CD45RA+) from total CD4 T cells (CD45hi and CD45lo), as this distinction is purely based on their activation status (total will include a mix of both. Naïve + memory). An additional marker such as CD197 would be necessary to distinguish the effector memory from the central memory, but with the gating strategy shown in Suppl Fig 3B,C, only naïve from memory cells can be distinguished. (based on CD45RA).

7. I believe it is important to analyze FOXP3 and RORgt expression side by side, as these transcription factors are generally mutually exclusive. It is difficult to visualize the induction of both transcription factors in a one-dimensional histogram plot. I also encourage the authors to include a live-dead marker in their gating strategy, as their FSC-SSC plot shows a significant number of dead cells that may interfere with their gating strategy in Supp Fig 4. Typically, when differentiating T helper cells, the process does not achieve 100% efficiency, and the authors should be able to clearly identify two populations, which is somewhat unclear in these histograms.

8. In lines 364-367, the authors discuss studies using Lck-Cre mice to generate conditional deletion of Trpm7 in T cells. However, it is unclear what the authors intend to convey in this context. In these studies, Lck-Cre was used to eliminate Trpm7 expression earlier in the maturation of T cells, specifically at the DN stage before the DP stage. T cells express LCK throughout their lifespan, even outside the thymus, but deletion of Trpm7 at the DN3 stage halts their maturation (PMID: 18974357). If Trpm7 is deleted later in the maturation process, it may prevent the arrest of T cell development at the DN stage and could potentially avoid the compensatory mechanisms that those clones may develop concerning their dependence on Mg²⁺ homeostasis. These are intriguing questions that should be addressed in future studies.

9. Please provide more detailed information regarding your staining of FOXP3. In the methods section, lines 519-522, there is no mention of fixation and permeabilization in your protocols. The same issue arises in lines 551-555, where you describe the intracellular staining of transcription factors.

Minor Comments:

1. The authors use GAPDH as a housekeeping control for their WB analysis. While this is acceptable given the short stimulation times (10-30 minutes), it's important to note that GAPDH experiences significant transcriptional and translational changes upon TCR stimulation due to the glycolytic switch. Actin is typically preferred as a housekeeping control in such cases.
2. Line 378, please change ...mice to "men". by ...mice to "humans".
3. In line 453, could you specify which antibodies were coated on the coverslips? Were they anti-CD3 and CD28? Please clarify.
4. Upon reviewing the references, I noticed that Reference 13 and 72 are the same. Ref 13 requires proper editing. Please make the necessary corrections.

Reviewer #3 (Comments to the Authors (Required)):

In the study "TRPM7 and magnesium orchestrate human CD4 T-cell activation and Differentiation" by Susanna Zierler and colleagues the authors investigate how a deletion of TRPM7 affects the activation and differentiation of human CD4+ T cells. In their study, the authors find that deletion of TRPM7 leads to distorted Mg²⁺ and Ca²⁺ signaling. The latter cumulates in reduced NFAT translocation, T cell activation, and an altered T helper (Th) cell differentiation with enhanced Foxp3 induced regulatory T cell but reduced Th17 polarization.

The study is well-designed and the topic is of interest to the scientific community. All conclusion derive from sufficient repeat experiments and solid datasets. Before publication I have a few minor suggestions:

- 1) In order to understand the underlying system for the readers it would be helpful if the authors could elaborate a bit on how sgRNA-mediated CRISPR/Cas9 gene editing of Jurkat cells was performed, f.e. how many guide RNAs were used to generate TRPM7-deficient Jurkat cells? How possible off-target effects by the guide RNAs were controlled? Such information could be

added to the material and methods section of the manuscript.

2) In Figure 4 the authors analyze the frequencies (%) of CD4⁺ T cells expressing the early activation markers CD69 and CD25 upon TRPM7 inhibition. Could the authors add the gate in the flow cytometry histograms on which the frequencies were determined? Could the authors add a quantitation of the MFI for CD69 and CD25 for the groups shown (=level of CD69 and CD25 protein expression)?

3) Could the authors check whether normalization of the CTRL samples in Figure panels 4A, 4F, 5B, 5D, 5Q and 5R are correct?

4) Did the authors measure TRPM7 protein expression in TRPM7-deficient Jurkat T cells?

Re: Life Science Alliance manuscript #LSA-2025-03357-T**Madlmayr & Hoelting et al. – TRPM7 and magnesium orchestrate human CD4 T-cell activation and differentiation****Reviewer comments**

We thank all reviewers for their valuable comments and suggestions, which led to the improvement and refinement of our manuscript. In response to the reviewers' comments, to incorporate new data and reanalyze previous data, we now rearranged the figures and corresponding results sections in the revised version of the manuscript. We are confident that the revised manuscript is now suitable for publication in *Life Science Alliance*.

Reviewer #1 (Comments to the Authors (Required)):

In this study, Madlmayr et al. investigate how TRPM7 affects the function of human T cells using CRISPR mediated deletion of TRPM7 (Jurkat T cell line) or a TRPM7 inhibitor (primary CD4 T cells). They show that deletion of TRPM7 in Jurkat cells inhibits TRPM7 currents, calcium influx, NFAT translocation and cell proliferation and viability. Similar results are obtained in primary CD4 T cells treated with the TRPM7 inhibitor NS8593. Intriguingly, anti-CD3 induced calcium influx is not abolished by NS8593, but calcium oscillations are. The authors link suppressed calcium oscillations to reduced production of IL-2 and expression of CD25 and CD69 by CD4 T cells. Mechanistically the most interesting part of the study are the data in figure 7, where the authors describe that NS8593 inhibits the phosphorylation of AKT and SMAD2, and blocks interaction TRPM7 with SMAD2. Moreover, NS8593 enhances the expression of Foxp3 but reduces that of ROR γ t during CD4 T cell differentiation in vitro, suggesting that TRPM7 regulates the balance of iTreg and Th17 cell fates. Overall the study is very interesting, well executed, the data are mostly convincing and the manuscript is well written. The results are consistent with the senior author's previous results in mouse T cells using mice expressing a kinase dead form of TRPM7, suggesting that the role of TRPM7 in mouse and human T cells is conserved. Most of my concerns are technical in nature, with one conceptual question regarding the role of Mg $^{2+}$ vs Ca $^{2+}$ in TRPM7 and T cell function.

Major comments:

1. Using ICP/MS to measure total cellular Mg $^{2+}$ in TRPM7 deficient Jurkat cells or Jurkat cells treated with NS8593, the authors show that Mg $^{2+}$ levels are reduced in the absence of TRPM7. This is in contrast to the original TRPM7 knockout paper in T cells by Jin et al. 2008 (Ref 13; minor comment: please fix format of Ref.13). How do the authors explain this discrepancy? And does this reduction of total cellular Mg $^{2+}$ matter for T cell function? I am asking because most subsequent analyses in this manuscript focus on the role of Ca $^{2+}$ signaling (and not Mg $^{2+}$) in TRPM7 deficient or inhibited T cells. This is also an important question because the authors measure Ca $^{2+}$ flux in TRPM7 deficient T cells and relate that finding to impaired T cell function, but they then add MgCl $_2$ to TRPM7 cells to rescue their function. This raises the question whether reduced Ca $^{2+}$ in TRPM7 deficient T cells is an epiphenomenon. (Another question is how does MgCl $_2$ enter TRPM7 deficient T cells).

Indeed, this is a very important point. We did reanalyse ICP-MS data based on the data analysis methods described in Jin et al. as shown in Figure 1R. As K $^{+}$ is an important signaling cation within cellular systems, it cannot be completely excluded that K $^{+}$ levels are unaffected in TRPM7 KO cells or upon its pharmacological blockade. First, we normalized K $^{+}$

levels in TRPM7 inhibited cells to S^- and P^{3-} (Fig 1R A and B, respectively), showing a significant reduction in total cellular K^+ levels upon NS8593 treatment. NS8593 was initially described as a SK-2 potassium channel blocker, effects on K^+ levels might be a compensatory effect by cells treated with NS8593. Both, normalization of P^{3-} to S^- and vice versa did not show any significant difference in Jurkat TRPM7 WT cells treated with NS8593 (Fig 1R C and D, respectively). Besides, the anions S^- or P^{3-} should be less involved in cellular signaling of cation channels. Normalization of cellular Mg^{2+} levels to S^- or P^{3-} did show a significant reduction of intracellular Mg^{2+} levels in Jurkat TRPM7 WT cells upon treatment with NS8593 (Fig 1R E and F, respectively). Similar to the pharmacological approach, normalization of P^{3-} to S^- and vice versa did not show any significant difference in Jurkat TRPM7 KO cells, showing the feasibility of this approach (Fig 1R G and H, respectively). Normalization of cellular Mg^{2+} levels to S^- or P^{3-} did show a significant reduction of intracellular Mg^{2+} levels in Jurkat TRPM7 KO cells, compared to WT (Fig 1R I and J, respectively). Normalization of Mg^{2+} levels to total K^+ or upon setting K^+ levels to 120 mM did not show any differences in intracellular Mg^{2+} levels between Jurkat TRPM7 WT and KO cells (Fig 1R K and L, respectively), though, as said, we refrain from applying a cation species (K^+) as basis for analysis here. In this regard, it would be interesting to analyze the data from Jin et al. with respect to normalization to non-cationic species like S^- or P^{3-} .

Changes of intracellular Mg^{2+} levels are important for numerous biological processes, as Mg^{2+} is associated to foster structural integrity of proteins, lipid membranes, nucleic acids, and its majority is complexed in the form of $Mg:ATP$ (Chubanov et al., 2016, Elife, PMID: 27991852; Liu et al., 2025, Nutrients, PMID: 40077788; Baaij et al., 2014, Pflugers Arch., PMID: 24413910). Thus, Mg^{2+} ions are critically involved in all aspects of cellular signaling, strongly supported by our current findings. How the observed reduction in Ca^{2+} signaling as a consequence of TRPM7 KO or inhibition is influenced by Mg^{2+} perturbation on a molecular level, is subject of ongoing investigations and beyond the scope of this manuscript. On the question of how Mg^{2+} enters the cell upon knockout or pharmacological blockade of TRPM7, one can still only speculate. Up to now the scientific community lacks a clear conclusion in this regard, and proteins including CNNMs and SLC41A transporters have been suggested to compensate for loss of TRPM7 (refs 62, 54 from the main manuscript). It will be interesting to follow up on these questions. Clearly, our data support our current understanding of TRPM7 ion channel as Mg^{2+} source in mammalian cells, as shown in various other cell lines (refs 14, 18, 20, 61 from the main manuscript). In the current version of the manuscript, these topics are discussed in lines 368-386.

Figure R1: Re-analysis of ICP-MS data.

A-B) Total cellular K⁺ levels normalized to S⁻ (A) and P³⁻ (B) in Jurkat TRPM7 WT cells treated with NS8593, compared to untreated Ctrl. C) Total P³⁻ levels normalized to S⁻ of Jurkat TRPM7 WT cells treated with NS8593, compared to Ctrl. D) Total S⁻ levels normalized to P³⁻ of Jurkat TRPM7 WT cells treated with NS8593, compared to Ctrl. E-F) Total cellular Mg²⁺ content normalized to cellular S⁻ levels (E) and P³⁻ levels (F) in Jurkat TRPM7 WT cells treated with NS8593, compared to Ctrl. G) Total P³⁻ levels normalized to S⁻ of Jurkat TRPM7 WT cells compared to KO cells. H) Total S⁻ levels normalized to P³⁻ of Jurkat TRPM7 WT cells compared to KO cells. I-J) Total cellular Mg²⁺ content normalized to cellular S⁻ levels (I) and P³⁻ levels (J) in Jurkat TRPM7 WT and KO cells. K-L) Total cellular Mg²⁺ content normalized to cellular K⁺ levels (K) and to K⁺ levels set to 120 mM as described in Jin et al. (L), in Jurkat TRPM7 WT and KO cells. All measurements are n=4. Statistics: Student's t test (A-F) and one-way ANOVA (G-L). ** P<0.005; *** P<0.0005, **** P<0.0001, n.s. – not significant. Data are mean ± SD.

2. The fact that TRPM7 deletion does not affect CD69 expression in Jurkat cells whereas treatment with NS8593 does reduce CD69 levels raises concerns about the specificity of NS8593 for TRPM7 inhibition. The authors somewhat mitigate these concerns by showing that a potential off-target of NS8593, the potassium channel SK2, does not affect T cell function when inhibited with apamin. But there may be many other off-targets of NS8593 besides SK2. A better experiment in my opinion, which is easy to do, would be to show that NS8593 does not further suppress the function of TRPM7 deficient Jurkat cells (CD69, proliferation etc).

We thank the reviewer for raising this point and the suggestion on additional experiments for monitoring off-target effects. Upon treatment of TRPM7 KO cells with NS8593, we did not observe any effects on cellular viability, proliferation or Ca²⁺ signaling. Additionally, we now performed the requested experiments on CD69 expression. Here we followed a suggestion from Reviewer 3 and performed MFI analysis, showing that CD69 expression is indeed significantly reduced in Jurkat TRPM7 KO clones as well as Jurkat TRPM7 WT cells treated with NS8593. Importantly, treatment of Jurkat TRPM7 KO cells with NS8593 did not further suppress CD69 expression levels. Altogether, these results suggest no further off-target effects of NS8593 in this regard (**Fig S3 E-H, and L-M**; manuscript lines: 161-167, 1179-1186 and 1187-1191).

3. I am not entirely convinced of the pAKT and pSMAD2 data in Figure 5C-I. The housekeeping control in C (GAPDH) shows great variations, and it is unclear how pAKT was normalized in D. The best normalization would be to total AKT levels. Similar concerns for pSMAD2 in E. The flow cytometry data for pSMAD2 in I show reduced phosphorylation, but that is not really apparent in the histogram in H.

We apologize for any lack of clarity regarding the quantification of Western blot data, we have now specified this more accurately in the materials and methods section (Manuscript line: 680). First, in the revised version of the manuscript, we quantified pAKT and pSMAD2 levels in basal state and upon stimulation, showing a stimulation-dependent increase in protein phosphorylation. We agree that quantification of phospho-levels to respective total protein levels would be best, however this was not always possible on primary T cells due to the limited sample material and donor to donor variations, possibly reflecting the variations seen in the previous draft. We therefore performed additional Western blot experiments where we quantified total protein levels (AKT, SMAD2/3) with respect to the same housekeeping controls. As this quantification did not show any differences, the quantification of phospho-signals to respective housekeeping controls is a valuable compromise (**Fig S5 A-F**, manuscript lines: 268-269, 278-279 and 1225-1234). To further strengthen our findings, similar to what we have shown for TRPM7-SMAD2 interactions, we performed proximity ligation experiments where we were able to show protein-protein interaction of TRPM7 with AKT1, which is significantly reduced upon TRPM7 inhibition (**Fig 5 C-D**, manuscript lines: 269-271, 423-426 and 1081-1084). To our knowledge, this is the first proof of direct TRPM7-AKT1 interaction in living cells.

Regarding the related question on flow cytometry data, changes in pSMAD2 levels upon 15 min stimulation with TGF β are small, still, the inhibition of TRPM7 with NS8593 led to a significant reduction in phosphorylation levels. To ensure better visualization, we now provide new graphs (**Fig 5G and K**). To additionally support the flow cytometry data, we performed several additional Western blot experiments including a new quantification of pSMAD2 levels (**Fig 5 E, F, I and H**; manuscript lines: 1086-1087 and 1090-1093). We believe that the new visualization in combination with additional experiments should satisfactorily address these concerns.

4. Along similar lines, I am not fully convinced that MgCl₂ and NS8593 affect ROR γ t levels in T cells. The flow data in Figure 5P do not show a robust difference and it is unclear how data were quantified in Q. Please also show flow data for ROR γ t expression in cell treated with NS8593 (panel R).

We appreciate the feedback regarding the gating strategy on ROR γ t expression. We followed a suggestion from reviewer 2 and included a live-dead stain in our gating strategy, now providing an entirely new set of experimental data. To eliminate potential drug effects in flow cytometry readout, we performed experiments on a Cytex Northern Lights flow analyzer, enabling spectral unmixing and thus the compensation for any drug/antibody spillover effects. We have now included and updated all FACS histograms and respective gating strategies in the related figures, showing the significant reduction of ROR γ t expression upon NS8593 treatment (**Fig 6 and Fig S6 and 7**).

Minor comments:

1. Mention in Figure 1A,F or its legend that TRPM7 currents were extracted at both +80 and -80 mV.

We now added this information (Manuscript lines: 960 and 968, respectively).

2. In Figure 2C and I, please show NFATc1 staining without DAPI. It is otherwise difficult to see NFATc1 location. For the quantification of NFATc1 location, it would be better to show ratios of nuclear to cytoplasmic NFATc1 per cell rather than intensities. Ditto in Figure 3M-P. The authors could add line scan traces as in 3M and O to Figure 2 to better visualize nuclear

translocation. In Figure 3M, the IF images do not show a discernable difference in NFATc1 location, whereas the line scan data do. Are these data derived from the same cell?

We appreciate the feedback of the reviewer, however due to the relatively big nucleus and small cytosol of Jurkat cells and T cells in general, we refrain to calculate nuclear to cytoplasmic ratios, as a proper discrimination of the two compartments might be difficult. Instead, we added line scan data for Jurkat cells in unstimulated state to visualize NFAT localization in comparison to stimulation data (**Fig 2 D and M**; Manuscript lines: 137, 985-988 and 999-1002).

3. In Figure 3C, D, I and J, what does "delta Ca²⁺" mean?

We thank the reviewer for raising this point; we described quantification of Ca²⁺ imaging data now in more detail in the materials and methods section (Manuscript line: 564-566).

4. For the data in Figures 3-5, the authors compare effects of TRPM7 inhibition on naive CD4 T cells and conventional CD4 T cells. There are no data in the manuscript describing the properties of these subsets. Which markers do these cells express, and have the authors analyzed these post isolation?

We have performed post isolation analyses for both populations and already showed the respective FACS plots in the SI file (**now Fig 4 A-B**) of the original submission. We now added a short statement about the properties of the analyzed populations to the manuscript (Manuscript line: 178-179).

5. In Figure 5B, I suggest to add statistical significance between cells treated with the same concentration of NS8593 in the presence or absence of MgCl₂.

We appreciate the feedback; to ensure proper readability we kept the original graph in the main figure and added the graph with the additional statistical comparisons in the supplementary material (**Fig S4L**).

Reviewer #2 (Comments to the Authors (Required)):

In the study titled "TRPM7 and magnesium orchestrate human CD4 T-cell activation and differentiation," Madlmayr et al. investigate the role of the channel-kinase TRPM7 in regulating Mg²⁺ homeostasis during the activation, differentiation, and function of human CD4 T cells. They utilize both pharmacological inhibition and genetic ablation of the protein. The authors primarily focus on human CD4 T cells, employing a widely accepted T cell leukemia cell line, Jurkats, as well as primary CD4⁺ T cells isolated from healthy donors, which broadens the scope of their research. The study is well-designed, and the methodologies used are appropriate. However, I have a few recommendations that could help the authors enhance the manuscript and present more accurate information. Overall, I believe the study is relevant, as it represents a significant advancement in understanding the role of TRPM7 in human immune cells, particularly since the mechanisms of Mg²⁺ transport and homeostasis are still debated. I encourage the authors to address these major points, as doing so could expand the scope of their manuscript and improve its suitability for publication in Life Science Alliance. Please find my comments outlined below:

Major points:

1. The sequence of the sgRNA used to target TRPM7 for generating Jurkat clones 1 and 2 must be provided, at least in the methods section. A Western blot indicating the KO of TRPM7 would be ideal; however, if this is not possible due to the lack of good antibodies, the

authors should either provide the sgRNA sequence or the indel introduced by the sgRNA (sequencing results). Currently, the only evidence for the KO of TRPM7 in Jurkat cells is the absence of TRPM7 currents observed through patch clamp electrophysiology. While this method is acceptable, it is important to note that the authors used different concentrations of Mg²⁺ (i.e., 6 mM) to maintain the TRPM7-KO Jurkats alive, and Mg²⁺ is known to inhibit TRPM7. An alternative way to demonstrate the KO of TRPM7 in Jurkats would be to challenge the cells with Naltriben, a proven antagonist of TRPM7 that potentiates TRPM7 function independent of Mg²⁺ concentration (PMID: 24633576, PMID: 37156763).

We appreciate the feedback on the TRPM7 KO validation. We now stated the sgRNA sequence in the materials and methods section (Manuscript line: 483-491). In addition, we performed Sanger Sequencing, RT-qPCR and Western Blot experiments to further strengthen the KO characterization, clearly showing the expected frameshift deletion, reduction in *TRPM7* transcript and absent protein levels in KO cells (Fig S1A-C). We discussed the suggestion to use Naltriben as a TRPM7 agonist to further validate TRPM7 KO cells, however we would like to note that activating TRPM7 in cells that do not express a functional channel is impossible. Nonetheless we performed additional proliferation experiments of TRPM7 KO cells treated with various concentrations of Naltriben. Our results show that Naltriben treatment does not affect Jurkat TRPM7 KO cell proliferation (Fig 2R A-B). Please also note that patch-clamp experiments were performed without Mg²⁺ present in buffer solutions, as outlined in the materials and methods section. All experiments on KO cells were performed in media or solutions containing standard concentrations of Mg²⁺ (0.4 mM), unless stated as Mg²⁺-reconstitution experiments (6.4 mM).

Figure R2: TRPM7 agonist Naltriben does not affect Jurkat TRPM7 KO viability or proliferation.

A) Cell counts and B) viability of proliferating TRPM7 WT and KO Jurkat clones in medium, with and without supplementation with 6 mM MgCl₂ or various Naltriben concentrations, n=2-3, measured in duplicates.

2. In Figures 2B, 2H, and Supplementary Figure 1G, the authors should subtract the baseline when calculating the area under the curve (AUC) of intracellular calcium concentration. Generally, increases in intracellular Ca²⁺ concentrations are nonlinear, which can lead to an overestimation of low intracellular concentrations. A double-log transformation is needed to linearize these values. By quantifying the changes using AUC without subtracting the baseline, the changes may appear small due to the higher baseline compared to the delta. Subtracting the baseline will provide a more accurate reflection of the differences in change.

We appreciate the feedback on Ca²⁺ imaging quantification. We blotted the original results (Fig 3R A-B and D-E) side by side with the recalculated AUC with subtracted baseline values (Fig 3R C and F). Since Ca²⁺ influx in TRPM7 KO cells or upon pharmacological blockade with NS8593 is minimal, some responses drop below baseline resulting in negative AUC values. Negative AUC values might be explained by bleaching events which are minimized by using a ratiometric dye, however they cannot be completely excluded. As both calculations produce very similar results, we decided to keep the original data calculation in

the revised version of the manuscript. The alternative calculations are displayed below in **Fig. R3**, showing a side-by-side comparison.

Figure 3: Re-analysis of Ca^{2+} imaging quantification.

A) Fura-2 based imaging of cytosolic Ca^{2+} concentration of Jurkat T cells. Passive store release was induced with 5 μ M thapsigargin at the indicated time point (arrow) of WT (black) and TRPM7 KO1 (red) and KO2 (orange) Jurkat T cells. Baseline values are indicated by the black line. B) Original quantification of the area under the curve (AUC) of respective curves shown in A. C) Recalculated quantification of the area under the curve (AUC) by subtraction of baseline values of respective curves shown in A. D) Fura-2 based imaging of cytosolic Ca^{2+} concentration of Jurkat T cells treated with NS8593 compared to untreated control. Passive store release was induced with 5 μ M thapsigargin at the indicated time point (arrow) of untreated controls (black) and TRPM7 NS (red). Baseline values are indicated by the black line. E) Original quantification of the area under the curve (AUC) of respective curves shown in D. F) Recalculated quantification of the area under the curve (AUC) by subtraction of baseline values of respective curves shown in D.

3. Are the calcium oscillations shown in Figures 3B, 3H, and Supplementary Figure 2F representative of a single cell or an average? It is difficult to visualize calcium oscillations from averaged cells, as they are not synchronized. Please clarify this more accurately in the figure legend.

We agree that visualization of Ca^{2+} oscillations is difficult when averaging cells as they are not synchronized. Thus, we blotted a representative trace of a single cell. We now specified this more clearly in the figure caption (**Fig 3, Fig S2**, Manuscript lines: 1016-1016 and 1022 respectively).

4. In lines 129-140, the authors discuss their results regarding CD69 expression with the genetic deletion of TRPM7 and pharmacologic inhibition of the channel, in light of their calcium-NFAT data. To my knowledge, CD69 and other activation markers in T cells, such as CD44, are calcium-NFAT independent. Specifically, CD69 is dependent on NF- κ B and AP-1, and therefore its expression should not be significantly affected by reduced calcium signaling. TCR stimulation activates multiple signaling pathways, including AP-1, NF- κ B, and calcium-NFAT, which may synergize; however, reduced calcium signaling shouldn't have a strong influence on CD69 expression. IL-2 and IL-2Ra are both fully dependent on NFAT,

making them good readouts for a lack of calcium-NFAT signaling in T cells. The fact that the inhibitor NS8593 shows a stronger inhibition of CD69 in WT cells may suggest that the inhibitor acts on an additional target other than TRPM7. One way to test this discrepancy would be to apply NS8593 to the TRPM7-KO Jurkats and analyse CD69 expression to confirm that the effect is on-target (in other words, no reduction in CD69 would be expected). We thank the reviewer for raising this point, a similar argument was made by reviewer 1. We repeated experiments on CD69 expression on Jurkat TRPM7 KO cells treated with NS8593. For data analysis we followed a valuable suggestion from reviewer 3 and performed MFI analysis, showing that CD69 expression is significantly reduced in Jurkat TRPM7 KO clones as well as Jurkat TRPM7 WT cells treated with NS8593 (Fig 2H-I, Q-R, Fig S2L-M and Fig S3 L-M). Treatment of NS8593 of Jurkat TRPM7 KO cells did not further suppress CD69 expression levels, suggesting no further off-target effects of NS8593.

5. How is the quantification of the Western blot in Figure 4C conducted? Are the authors normalizing to GAPDH or to the control at the 0-time point? GAPDH is not the best housekeeping for normalizing transcriptional and protein levels in T cells due to the strong glycolytic switch that occurs upon TCR stimulation. Although the time points used in this study are short and should not significantly affect overall GAPDH expression, it is important to note that activation can trigger a changes in GAPDH levels. Normalizing analysed-AKT (Ser473) to total AKT would be a better approach.

A similar concern was raised by Reviewer 1 (point 3): We apologize for any lack of clarity regarding the quantification of Western blot data, we specified this in more detail in the materials and methods section (Manuscript lines: 680).

In the revised version of the manuscript, we now quantified pAKT and pSMAD2 levels in basal state and upon stimulation, showing a stimulation-dependent increase in protein phosphorylation. We agree that quantification of phospho-levels to respective total protein levels would be best, however this was not always possible on primary T cells due to the limited sample material and donor to donor variations, possibly reflecting the variations seen in the previous draft. We therefore performed additional Western blot experiments where we quantified total protein levels (AKT, SMAD2/3) with respect to the same housekeeping controls. As this quantification did not show any differences, the quantification of phospho-signals to respective housekeeping controls is a valuable compromise (Fig S5 A-F, manuscript lines:, 268-269, 278-279 and 1225-1234).

6. The authors use the terms “naïve CD4 T cells” and “conventional CD4 T cells” to distinguish their cells based on the CD45RA expression. In principle, all naïve T cells are conventional T cells as they have not differentiated towards the regulatory subset. I will be more cautious with that terminology and simply use naïve CD4 T cells (CD45RA+) from total CD4 T cells (CD45hi and CD45lo), as this distinction is purely based on their activation status (total will include a mix of both. Naïve + memory). An additional marker such as CD197 would be necessary to distinguish the effector memory from the central memory, but with the gating strategy shown in Suppl Fig 3B,C, only naïve from memory cells can be distinguished. (based on CD45RA).

We thank the reviewer for raising this important point and apologize for unclarities in terminology. In our study we investigated the naïve CD4⁺ T-cell population in comparison with the total CD4⁺ T-cell population, and did not further discriminate for effector and/ or central memory T cells, and thus did not stain for any other markers in our post isolation control measurements. We have clarified and amended the terminology throughout the revised version of the manuscript.

7. I believe it is important to analyze FOXP3 and ROR γ t expression side by side, as these transcription factors are generally mutually exclusive. It is difficult to visualize the induction of both transcription factors in a one-dimensional histogram plot. I also encourage the authors to include a live-dead marker in their gating strategy, as their FSC-SSC plot shows a significant number of dead cells that may interfere with their gating strategy in Supp Fig 4. Typically, when differentiating T helper cells, the process does not achieve 100% efficiency, and the authors should be able to clearly identify two populations, which is somewhat unclear in these histograms.

We highly appreciate the feedback on our T-cell differentiation experiments. We repeated all differentiation experiments including a live-dead stain and performed the new flow measurements on a Cytex Northern Lights analyzer, allowing spectral unmixing and thus to compensate more accurately for any drug/ antibody spillover effects (materials and methods section, lines 644-647 and 659-611). We agree with the reviewer, our T-cell differentiation did not achieve 100% efficiency especially upon polarization into the T_H17 lineage, which is generally known to be challenging in human T cells (Revu et al., 2018, CellRep., PMID: 29514093). In addition to the suggestion of the reviewer, we analyzed percentages of CCR6 expressing cells after gating for cellular viability, and gated on ROR γ t expression in CCR6⁺ live cells, for T_H17 experiments. Interestingly, Soongsathitanon et al. (2024, Heliyon, PMID: 38807879) recently showed that expression of ROR γ t and FOXP3 are not always mutually exclusive, especially in human T_H17 cells and upon in vitro differentiation. Accordingly, when we performed experiments on ROR γ t side by side with FOXP3 staining, we did not gain conclusive results, thus we decided to plot ROR γ t expression in our newly designed flow panel upon compensation of potential drug effects. Our data confirm the significant reduction of ROR γ t levels upon NS8593 treatment (see results **Fig 6J-O and S7A-F**). For T_{reg} differentiation experiments we additionally blotted CD45RA status and percentages of CD25⁺CD127^{lo} cells to strengthen and support our data (see **Fig 6B-I and S6A-F**). Repeating these experiments, including the live-dead stain as well as by blotting additional expression markers, we majorly improved our manuscript and hope to have adequately addressed the concerns of the reviewer.

8. In lines 364-367, the authors discuss studies using Lck-Cre mice to generate conditional deletion of Trpm7 in T cells. However, it is unclear what the authors intend to convey in this context. In these studies, Lck-Cre was used to eliminate Trpm7 expression earlier in the maturation of T cells, specifically at the DN stage before the DP stage. T cells express LCK throughout their lifespan, even outside the thymus, but deletion of Trpm7 at the DN3 stage halts their maturation (PMID: 18974357). If Trpm7 is deleted later in the maturation process, it may prevent the arrest of T cell development at the DN stage and could potentially avoid the compensatory mechanisms that those clones may develop concerning their dependence on Mg²⁺ homeostasis. These are intriguing questions that should be addressed in future studies.

With the statement made in the original manuscript, we intended to point out that since Trmp7 deletion in Lck-Cre mice leads to a developmental arrest keeping T lymphocytes in the double negative state, it is difficult to compare data derived from these murine cells with our pharmacological approach on fully developed human CD4 T cells. To clarify this point, we rewrote this part in the discussion (manuscript lines: 456-460)

9. Please provide more detailed information regarding your staining of FOXP3. In the methods section, lines 519-522, there is no mention of fixation and permeabilization in your

protocols. The same issue arises in lines 551-555, where you describe the intracellular staining of transcription factors.

Thank you for mentioning this point: we clarified the fixation and permeabilization as well as staining procedure in more detail for both, FOXP3 staining (Manuscript line: 523641-643) as well as staining of transcription factors (Manuscript line: 692-694).

Minor Comments:

1. The authors use GAPDH as a housekeeping control for their WB analysis. While this is acceptable given the short stimulation times (10-30 minutes), it's important to note that GAPDH experiences significant transcriptional and translational changes upon TCR stimulation due to the glycolytic switch. Actin is typically preferred as a housekeeping control in such cases.

We appreciate the feedback regarding potential changes in GAPDH expression upon TCR stimulation. We selected GAPDH as a housekeeping control as TRPM7 has not been shown to modulate GAPDH, unlike other frequently used housekeeping genes such as Histone H3 (Qiao et al., 2021, NatCommun, PMID: 34001887), myosin II heavy chain (Clark et al., 2008, FEBS Lett., PMID: 18675813) or RHOA (Voringer et al., 2020, Oncogene, PMID: 31844251), of which the latter two are involved in actin dynamics. Since donor-to-donor variability provides a variation of T-cell responses, we now performed additional blots and provide a reanalysis of the data, also including total-protein controls (AKT, SMAD2/3) versus GAPDH (Fig 5A-F, manuscript lines: 268-269, 278-279 and 1225-1234).

2. Line 378, please change ...mice to "men". by ...mice to "humans".

We changed the wording accordingly.

3. In line 453, could you specify which antibodies were coated on the coverslips? Were they anti-CD3 and CD28? Please clarify.

Indeed, as outlined in the figure caption (Fig 3B and H), T cells were activated with α CD3/ α CD28 antibodies. We have additionally clarified this in the materials and methods section (Manuscript line: 560).

4. Upon reviewing the references, I noticed that Reference 13 and 72 are the same. Ref 13 requires proper editing. Please make the necessary corrections.

We apologize for the inconsistent referencing to reference 13/72. We made the necessary corrections in the revised version of the manuscript.

Reviewer #3 (Comments to the Authors (Required)):

In the study "TRPM7 and magnesium orchestrate human CD4 T-cell activation and Differentiation" by Susanna Zierler and colleagues the authors investigate how a deletion of TRPM7 affects the activation and differentiation of human CD4+ T cells. In their study, the authors find that deletion of TRPM7 leads to distorted Mg²⁺ and Ca²⁺ signaling. The latter cumulates in reduced NFAT translocation, T cell activation, and an altered T helper (Th) cell differentiation with enhanced Foxp3 induced regulatory T cell but reduced Th17 polarization.

The study is well-designed and the topic is of interest to the scientific community. All conclusion derive from sufficient repeat experiments and solid datasets. Before publication I have a few minor suggestions:

1) In order to understand the underlying system for the readers it would be helpful if the authors could elaborate a bit on how sgRNA-mediated CRISPR/Cas9 gene editing of Jurkat cells was performed, f.e. how many guide RNAs were used to generate TRPM7-deficient Jurkat cells? How possible off-target effects by the guide RNAs were controlled? Such information could be added to the material and methods section of the manuscript.

We thank the reviewer for raising this point: We added the sgRNA sequence and more information in the materials and methods section (Manuscript line: 483-491). As outlined, the Jurkat TRPM7 KO clones were generated by ThermoFisher. The entire procedure (design of sgRNA sequence, editing, single cells seeding and clonal expansion) was performed by the company in close collaboration with us. In addition, we now added Sanger sequencing results, Western Blot and RT-qPCR validation of the KO clones, showing the expected frameshift mutation in KO sequences, reduced *TRPM7* mRNA transcripts and absent TRPM7 protein expression in the KO clones (**Fig S1A-C**). We addressed the question on potential off-target effects by validating a second, individual KO clone side by side, these data were shown in the supplementary material (Fig S1 of the original submission). As both TRPM7 KO clones as well as our pharmacological approach using NS8593 show similar results, the likelihood of assessing off-target effects is minimal.

2) In Figure 4 the authors analyze the frequencies (%) of CD4+ T cells expressing the early activation markers CD69 and CD25 upon TRPM7 inhibition. Could the authors add the gate in the flow cytometry histograms on which the frequencies were determined? Could the authors add a quantitation of the MFI for CD69 and CD25 for the groups shown (=level of CD69 and CD25 protein expression)?

We thank the reviewer for pointing out the incomplete gating strategy. For clarification, we have added the gates in the flow cytometry data. It is commonly accepted practice to analyze bi-shaped populations in flow histograms by extraction of percentage values, while a total shift of the peak shall be best analyzed by MFI evaluation (Sino Biological Inc.: Flow cytometry data analysis). It is not advised to analyze a bi-phasic peak by MFI extraction, as no meaningful data would arise. We adjusted our FACS analysis where necessary (**Fig 2 H-I, Q-R, Fig S2L-M and S3J-M**). Since CD69 and CD25 expression in primary human T cells show a bi-phasic population (**Fig. 4B, D, G, I, L and N**), we kept the original analysis and display the extraction of percentages.

3) Could the authors check whether normalization of the CTRL samples in Figure panels 4A, 4F, 5B, 5D, 5Q and 5R are correct?

We appreciate the feedback of the reviewer. We recalculated the normalization and set the control to 100%.

4) Did the authors measure TRPM7 protein expression in TRPM7-deficient Jurkat T cells?

We thank the reviewer for raising this concern. For further Jurkat TRPM7 KO validation we have performed additional Western blot experiments and added this to the revised version of the manuscript, alongside a depiction of Sanger sequencing traces of WT and KO cells, as well as a RT-qPCR analysis of *TRPM7* mRNA transcripts (**Fig S1A-C**, manuscript lines: 1142-1146). These collected data confirm the KO status in the TRPM7-targeted clones.

November 3, 2025

RE: Life Science Alliance Manuscript #LSA-2025-03357-TR

Prof. Susanna Zierler
Johannes Kepler University of Linz
Institute of Pharmacology
MED Campus I, Krankenhausstr. 5
Linz 4020
Austria

Dear Dr. Zierler,

Thank you for submitting your revised manuscript entitled "TRPM7 and magnesium orchestrate human CD4 T-cell activation and differentiation". As you will see, reviewers are satisfied with no further requests. We would be happy to publish your paper in Life Science Alliance pending final revisions necessary to meet our formatting guidelines.

- Please add the X and Bluesky handles of your host institute/organization, as well as your own and/or one of the authors, in our system.
- Please add a Data Availability section, placed after the Materials & Methods section. Please consult our guidelines at <https://www.life-science-alliance.org/manuscript-prep#format>.
- Please consult our manuscript preparation guidelines <https://www.life-science-alliance.org/manuscript-prep> and make sure your manuscript sections are in the correct order.
- Please be sure that the names of the authors listed in the authors' contribution section are correctly written and match the system.
- Please add your main and supplementary figure legends to the main manuscript text after the references section.
- We encourage you to revise the figure legends for figures 6 and S3 such that the figure panels are introduced in alphabetical order.

LSA now encourages authors to provide a 30-60 second video where the study is briefly explained. We will use these videos on social media to promote the published paper and the presenting author (for examples, see <https://docs.google.com/document/d/1-UWCfbE4pGcDdcgzcmiuJl2XMBJnxKYeqRvLLrLSo8s/edit?usp=sharing>). Corresponding or first-authors are welcome to submit the video. Please submit only one video per manuscript. The video can be emailed to contact@life-science-alliance.org

A. FINAL FILES:

B. MANUSCRIPT ORGANIZATION AND FORMATTING:

Thank you for your attention to these final processing requirements. Please revise and format the manuscript and upload materials as soon as you are able.

Sincerely,

Reviewer #2 (Comments to the Authors (Required)):

The authors have effectively addressed all my major and minor points raised during the revision. Congratulations on a great piece of work!

Reviewer #3 (Comments to the Authors (Required)):

The authors have addressed all comments in detail and need to be congratulated to the nice work.

November 16, 2025

RE: Life Science Alliance Manuscript #LSA-2025-03357-TRR

Prof. Susanna Zierler
Johannes Kepler University of Linz
Institute of Pharmacology
MED Campus I, Krankenhausstr. 5
Linz 4020
Austria

Dear Dr. Zierler,

Thank you for submitting your Research Article entitled "TRPM7 and magnesium orchestrate human CD4 T-cell activation and differentiation". It is a pleasure to let you know that your manuscript is now accepted for publication in Life Science Alliance. Congratulations on this interesting work.

Your manuscript will now progress through copyediting and proofing. When reviewer proofs, please correct the Data Availability statement which would cause reader confusion as currently phrased. Please also indicate the availability of underlying data in addition to materials. It is journal policy that authors provide original data upon request.

DISTRIBUTION OF MATERIALS:

Again, congratulations on a very nice paper. I hope you found the review process to be constructive and are pleased with how the manuscript was handled editorially. We look forward to future exciting submissions from your lab.

Sincerely,
